# Efficient Multi-modal Large Language Models via Progressive Consistency Distillation

**Zichen Wen**[1,2]    **Shaobo Wang**[1]    **Yufa Zhou**[3]    **Junyuan Zhang**[4]    **Qintong Zhang**[5]
**Yifeng Gao**[1]    **Zhaorun Chen**[6]    **Bin Wang**[2]    **Weijia Li**[7,2]
**Conghui He**[2*]  **Linfeng Zhang**[1*]

[1]EPIC Lab, Shanghai Jiao Tong University    [2]Shanghai AI Laboratory
[3]Duke University    [4]The University of Hong Kong
[5]Peking University    [6]University of Chicago    [7]Sun Yat-sen University
heconghui@pjlab.org.cn, zhanglinfeng@sjtu.edu.cn
 **Code:** https://github.com/ZichenWen1/EPIC

## Abstract

Visual tokens consume substantial computational resources in multi-modal large models (MLLMs), significantly compromising their efficiency. Recent works have attempted to improve efficiency by compressing visual tokens during training, either through modifications to model components or by introducing additional parameters. However, they often overlook the increased learning difficulty caused by such compression, as the model's parameter space struggles to quickly adapt to the substantial perturbations in the feature space induced by token compression. In this work, we propose to develop **E**fficient MLLMs via **P**rogress**I**ve **C**onsistency Distillation (EPIC), a progressive learning framework. Specifically, by decomposing the feature space perturbations introduced by token compression along the token-wise and layer-wise dimensions, we introduce token consistency distillation and layer consistency distillation, respectively, aiming to reduce the training difficulty by leveraging guidance from a teacher model and following a progressive learning trajectory. Extensive experiments demonstrate the superior effectiveness, robustness, and generalization capabilities of our proposed framework.

## 1 Introduction

Multi-modal large language models (MLLMs) [32, 81, 39, 42] equip large language models (LLMs) [54, 23] with the ability to understand visual information, exhibiting remarkable capabilities across a diverse range of multi-modal tasks, including image captioning, visual question answering (VQA), video understanding [61], and multi-modal reasoning [62].

However, unlike LLMs [54, 2, 66, 59], which only need to process a small number of information-dense text tokens [48], the introduction of substantial visual tokens in MLLMs [3, 4] presents significant computational challenges. This issue becomes particularly pronounced when handling high-resolution images [35] or multi-frame videos [53].

To address this issue, a natural approach is to reduce visual tokens [44, 65, 6, 67]. Recent advances have introduced various token compression techniques to eliminate vision tokens in a training-free approach [78, 63]. Most of them adopt either a token importance-based strategy [7, 24] or a token redundancy-based strategy [57, 52]. Although the aforementioned non-parametric methods avoid additional training costs, they inevitably incur significant performance degradation. To achieve

---

*Corresponding authors.

39th Conference on Neural Information Processing Systems (NeurIPS 2025).

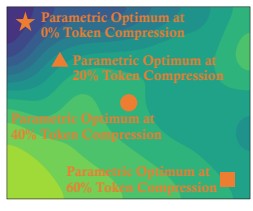
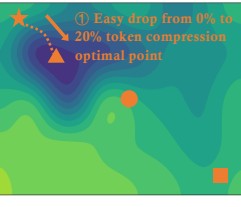
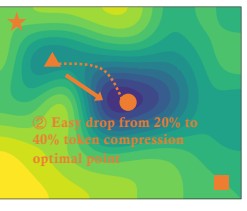
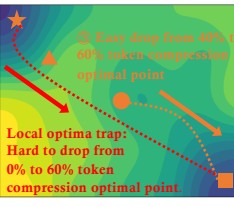

| (a) Loss Landscape of Compression Ratio = 0% | (b) Loss Landscape of Compression Ratio = 20% | (c) Loss Landscape of Compression Ratio = 40% | (d) Loss Landscape of Compression Ratio = 60% |

Figure 1: Progressive Consistency Distillation vs. Direct Training. Each subplot shows the loss landscape under the corresponding token compression ratio, with the **optimum** indicated. Our method reaches the objective via **progressive** learning trajectories, while **direct** training remains challenging.

a better performance-efficiency trade-off, recent training-aware token compression methods have attracted significant attention [5, 14]. Beyond early approaches [32] that employed Q-former, MQT-LLaVA [26] proposes a more flexible Q-former capable of dynamically encoding visual information into variable-length visual tokens. [33] proposes TokenPacker, a coarse-to-fine visual projector that progressively refines coarse tokens with visual information. In addition to refining the model architecture and upgrading model components, VoCo-LLaMA [70] and LLaVA-Mini [76] build upon observations of the transfer of visual information within language models, attempting to transfer visual information into a small set of VoCo tokens via attention modification, or into textual tokens with a transformer-based pre-fusion module.

Although these methods improve model efficiency while largely preserving performance, their improvements primarily come from architectural enhancements or newly introduced modules, often overlooking the learning challenges posed by token compression during training. As illustrated in Figure 1, when token compression is applied during training, the distribution of the compressed token sequence inevitably differs from that of the full token set. This discrepancy can be regarded as a perturbation in the feature space, which shifts the model's optimal point in the parameter space. The higher the compression ratio, the greater the perturbation introduced, and consequently, the further the optimal point drifts. The goal of training-aware token compression is to guide the model to progressively adapt from the original optimum (under full tokens) to a new optimum corresponding to the compressed token distribution. Figure 1 (d) shows that directly training a model with compressed tokens often leads to suboptimal solutions, as the optimization process can easily get trapped in poor local minima, making it difficult to reach the desired optimum under heavy compression.

In this work, we propose a progressive consistency distillation learning framework EPIC tailored for token compression, where a single MLLM simultaneously acts as both teacher and student through weight sharing. From a token-wise perspective, we introduce **Token Consistency Distillation (TCD)**. At the early stages of training, both the teacher and student models adopt a very low token compression ratio, indicating a relatively easy learning task without significant optimal point drifts (as shown in Fig. 1 (b)). As training progresses, the compression becomes increasingly aggressive, forming a progressive learning trajectory (see Fig. 1 (b) to (d)). Although the final optimal point under heavy compression remains far from the initial optimum, each intermediate optimum along the trajectory is relatively close to its predecessor, making each transition more manageable and easier to optimize. Moreover, the teacher consistently uses a slightly lower compression ratio than the student, introducing a compression ratio gap between them. We argue that when the gap is too large, the student may struggle to benefit effectively from the teacher's guid-

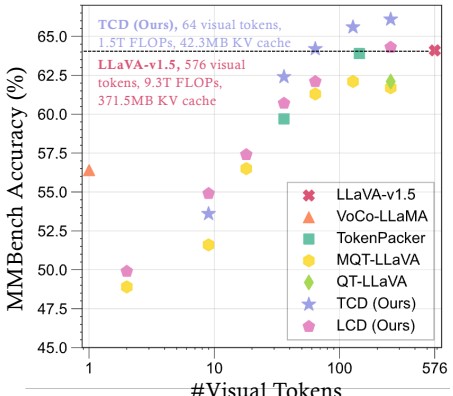

Figure 2: MMBench accuracy vs. number of visual tokens for various methods. TCD (Ours) and LCD (Ours) achieve competitive accuracy with far fewer tokens, lower FLOPs, and smaller KV cache compared to LLaVA-v1.5, highlighting its efficiency.

ance [20]. Therefore, the compression ratio gap is also designed to follow a progressive learning strategy, gradually increasing over time to ease the learning process. From a layer-wise perspective, we introduce **Layer Consistency Distillation (LCD)**. Based on observations from prior work [7], the significance of visual tokens diminishes notably in deeper layers, suggesting that compressing tokens

at these layers has minimal impact on the model's feature space and output. Therefore, in LCD, the token compression is progressively shifted from deeper to shallower layers as training progresses, implicitly following an easy-to-hard learning paradigm. Meanwhile, a compression gap between the teacher and the student is maintained to encourage effective guidance from the teacher.

Our primary contribution lies in proposing EPIC, a progressive consistency distillation learning framework, which demonstrates compatibility with diverse token compression techniques. Within this framework, we introduce Token Consistency Distillation and Layer Consistency Distillation from token-wise and layer-wise dimensions, respectively. This approach enables the training of robust and highly generalizable models through a progressive learning strategy without requiring modifications to the model architecture. Compared to prior approaches, comprehensive experiments validate the superior effectiveness, robustness, and generalization capabilities of our proposed framework.

## 2 Related Works

**Multi-modal Large Language Models.** Multi-modal Large Language Models (MLLMs), pioneered by [41, 81, 15, 71, 8] have successfully showcased promising results on a wide variety of vision-language perception and reasoning tasks. Existing MLLMs typically employ a pre-trained vision encoder (*e.g.*, CLIP [49] and SigLIP [73, 55]) to extract visual features, which are then projected into the LLM's input space via a visual projector (*e.g.*, MLP and Q-former [32]), enabling the model to process both visual embeddings and user instructions for multimodal understanding and response generation. Recent studies [75, 74] have highlighted the limitations of MLLMs in fine-grained visual perception tasks. To this end, more advanced MLLMs have attempted to increase the number of encoded visual tokens by employing dynamic resolutions [11, 10, 40, 31] or arbitrary resolutions [58, 4] to process high-resolution images, thereby enhancing their performance in visual understanding tasks. However, due to the quadratic complexity of the attention mechanism [56], the resulting longer token sequences pose significant challenges to both inference speed and memory usage.

**Visual Token Compression.** Visual tokens typically outnumber text tokens by orders of magnitude while containing greater spatial redundancy than information-dense text [48]. Recent work has explored both training-free [24, 79, 80, 43, 77] and training-aware [5, 13, 3, 9, 60] compression approaches, with the latter showing superior performance potential despite requiring additional training [63]. Training-free methods generally follow two paradigms: importance-based strategies like FastV [7] and SparseVLM [78] that use attention scores, and redundancy-based approaches such as DART [64] and G-Prune [28] that assess token similarity. Early training-aware work focused on parameter-free strategies, including LLaVA-PruMerge's attention-based merging [50] and VoCo-LLaMA's compressed VoCo tokens [70]. Subsequent research explored architectural modifications through lightweight component replacements [16, 19, 34]. More advanced approaches include MQT-LLaVA's dynamic Q-former for variable-length token encoding [26] and TokenPacker's coarse-to-fine iterative condensation [33]. Recent methods like LLaVA-Mini [76] achieve near-lossless token compression through extra auxiliary modules, though they often overlook the training challenges introduced by feature space perturbations during token compression.

## 3 Methodology

### 3.1 Preliminary

**Multi-modal Large Language Models (MLLMs).** An MLLM typically consists of three modules: a *visual encoder* VE, a *modality projector* (*e.g.*, MLP), and a *language model* LM. Given image $\mathcal{I}$, the visual encoder extracts patch-level features $\mathbf{z}_v \in \mathbb{R}^{N \times d_v}$, projected into visual tokens $\mathbf{e}_v \in \mathbb{R}^{N \times d_h}$:

$$\mathbf{e}_v = \mathrm{MLP}(\mathrm{VE}(\mathcal{I})), \tag{1}$$

where $N$ is the number of image patches, $d_v$ is the encoder output dimension, and $d_h$ is the LM hidden size. Meanwhile, a text prompt $\mathcal{P}$ is tokenized and embedded into a sequence of text tokens $\mathbf{e}_t \in \mathbb{R}^{L \times d_h}$. The visual and text tokens are then concatenated to form the full input sequence:

$$\mathbf{x} = [\mathbf{e}_v; \mathbf{e}_t] \in \mathbb{R}^{(N+L) \times d_h}. \tag{2}$$

Positional embeddings are added to $\mathbf{x}$ to encode spatial and sequential structure. The language model then autoregressively generates output tokens $y_i \in \mathcal{V}$ (with vocabulary $\mathcal{V}$) one token at a time:

$$y_i = \mathrm{LM}(\mathbf{x}, y_{<i}), \quad \text{for } i = 0, 1, 2, \dots \tag{3}$$

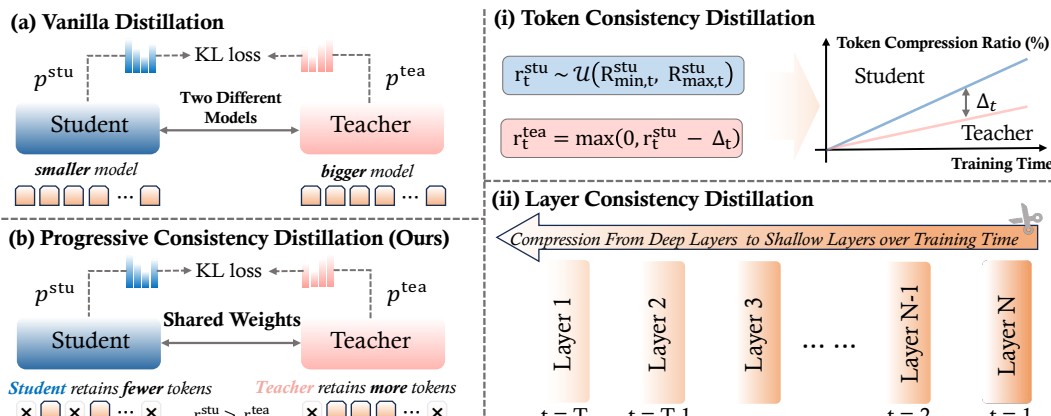

Figure 3: **An overview of Progressive Consistency Distillation.** (i) Token Consistency Distillation progressively increases token compression ratio over time. (ii) Layer Consistency Distillation shifts token compression from deep to shallow layers, promoting layer-wise consistency during training.

where $y_{<i} := \{y_0, y_1, \dots, y_{i-1}\}$ denotes the previously generated tokens.

**Progressive Consistency Distillation Learning.** Training MLLMs under aggressive token compression introduces significant feature space perturbations that hinder convergence. To address this, we propose *Progressive Consistency Distillation Learning* (PCDL), which gradually increases compression difficulty and facilitates the model's progressive convergence to the final objective (see Fig. 1). Our **Token Consistency Distillation (TCD)** progressively increases the token compression ratio over training steps from a token-wise perspective, while **Layer Consistency Distillation (LCD)** initially applies token compression in deeper layers (minimal impact [7, 76]) before gradually shifting to shallower layers, performing progressive learning in a layer-wise manner. Both employ *Consistency Distillation with Shared Weights*, where teachers with slightly lower compression ratios (*e.g.*, 5% less) provide manageable guidance [20], progressively transitioning to stronger teachers (*e.g.*, 10% lower ratio) for staged mentorship.

### 3.2  Theoretical Intuition: A 1D Prototype for Progressive Consistency Distillation Learning

**Scalar center path.** To provide intuition for *Progressive Consistency Distillation Learning* (PCDL), we introduce a one-dimensional prototype where model predictions are scalar values $\theta \in \mathbb{R}$, and each target is given by a compression-dependent center $c_r \in \mathbb{R}$ for compression ratio $r \in [0, r_{\max}]$. We assume that the mapping $c : [0, r_{\max}] \to \mathbb{R}$ satisfies:

**(S1)** *Monotonicity:* $c$ is differentiable with $c_r' \geq 0$ for all $r$.

**(S2)** *Lipschitz slope:* There exists $\gamma > 0$ such that $|c_r'| \leq \gamma$, and $|c_{r_1} - c_{r_2}| \leq \gamma |r_1 - r_2|$ for all $r_1, r_2$.

**(S3)** *Convexity:* $c$ is convex, i.e., $c''(r) \geq 0$.

**Two quadratic objectives.** We compare two learning objectives:

$$\mathcal{L}_{\text{dir}}(r, \theta) = \tfrac{1}{2}(\theta - c_r)^2, \qquad \mathcal{L}_{\text{prog}}(r, \theta) = \tfrac{1}{2}(\theta - c_r)^2 + \tfrac{\lambda}{2}(\theta - c_{r-\Delta})^2,$$

with constants $0 < \lambda < 1$, $0 < \Delta \leq r_{\max}$. The second term acts as a regularizer pulls the prediction toward a slightly less compressed teacher target, mimicking the KL-based distillation loss in EPIC.

**Analogy to EPIC.** In EPIC, the teacher operates at a slightly lower compression ratio $(r - \Delta)$ and provides smoother targets via KL consistency. The scalar prototype captures this: $\mathcal{L}_{\text{dir}}$ corresponds to direct supervision from the current target $c_r$, while $\mathcal{L}_{\text{prog}}$ encodes progressive distillation by incorporating a past (easier) target $c_{r-\Delta}$. As in EPIC, the regularizer improves training stability by reducing sensitivity to abrupt changes in the input space induced by token compression.

**Exact minimizers.** Both objectives are strongly convex in $\theta$ with closed-form solutions:

$$\theta_r^{\text{dir}} = c_r, \qquad \theta_r^{\text{prog}} = \frac{c_r + \lambda \, c_{r-\Delta}}{1 + \lambda}.$$

**Schedule and path length.** Let $0 = r_0 < r_1 < \cdots < r_T = r_{\max}$ denote any compression schedule. The total variation of a path $\{x_t\}$ is defined as:

$$\mathrm{TV}(\{x_t\}) := \sum_{t=0}^{T-1} |x_{t+1} - x_t|.$$

We show below that the progressive path yields strictly shorter total variation, reflecting the smoother optimization trajectory induced by PCDL. We delay the full proof to Appendix B.

**Theorem 1** (Scalar path gain, Proof in Appendix B). *Under assumptions (S1)–(S3), the total variation of the progressive path is strictly smaller:*

$$\mathrm{TV}(\{\theta_{r_t}^{\mathrm{dir}}\}) \le \gamma r_{\max},$$

$$\mathrm{TV}(\{\theta_{r_t}^{\mathrm{prog}}\}) \le \frac{1 + \lambda\kappa}{1 + \lambda} \cdot \mathrm{TV}(\{\theta_{r_t}^{\mathrm{dir}}\}) < \mathrm{TV}(\{\theta_{r_t}^{\mathrm{dir}}\}),$$

$$\textit{where} \qquad \kappa := \sup_{r \in [\Delta,\, r_{\max} - \Delta]} \frac{c(r) - c(r - \Delta)}{c(r + \Delta) - c(r)} \in [0, 1).$$

### 3.3 Token Consistency Distillation

We consider a single MLLM $f_\theta$ that plays both teacher and student roles by sharing parameters $\theta$. Let $\mathcal{I}$ denote the input image and $\mathcal{P}$ denote the language prompt. Let $C(\mathcal{I}, r, \ell)$ be a token compression operator[1] that, at layer $\ell$, compresses the visual tokens extracted from $\mathcal{I}$ with ratio $r \in [0, 1]$, *i.e.*, retaining only a fraction $1 - r$ of visual tokens. Let $t \in \{0, \ldots, T\}$ index training steps.

At iteration $t$, we sample a student compression ratio from a gradually shifting uniform distribution:

$$r_t^{\mathsf{stu}} \sim \mathcal{U}\big(R_{\min,t}^{\mathsf{stu}},\ R_{\max,t}^{\mathsf{stu}}\big), \tag{4}$$

where $R_{\max,t}^{\mathsf{stu}}$ linearly increases from $R_{\max,0}^{\mathsf{stu}} = \epsilon$ (*e.g.*, 5%) to $R_{\max,T}^{\mathsf{stu}} = R_{\max}$ (*e.g.*, 90%) with training steps, and $R_{\min,t}^{\mathsf{stu}}$ grows more slowly from 0% to at most 50% as training progresses. This creates an easy-to-hard curriculum for the student model by gradually narrowing and shifting the sampling range towards more aggressive compression.

We then define the teacher model's compression ratio as

$$r_t^{\mathsf{tea}} = \max\big(0,\ r_t^{\mathsf{stu}} - \Delta_t\big), \tag{5}$$

where the compression gap $\Delta_t$ also increases gradually from $\Delta_0 = \delta_{\min}$ to $\Delta_T = \delta_{\max}$ (*e.g.*, up to 30%), ensuring that the teacher consistently sees slightly less compressed inputs than the student.

Concretely, we form two forward passes through the shared model $f_\theta$:

$$\mathbf{h}^{\mathsf{tea}} = f_\theta\big(C(\mathcal{I}, r_t^{\mathsf{tea}}, \ell); \mathcal{P}\big); \quad \mathbf{h}^{\mathsf{stu}} = f_\theta\big(C(\mathcal{I}, r_t^{\mathsf{stu}}, \ell); \mathcal{P}\big), \tag{6}$$

where both token compressions occur at the fixed Transformer layer $\ell$.

We incorporate the Token Consistency Distillation (TCD) loss as an auxiliary objective alongside the main supervised fine-tuning (SFT) loss. Specifically, the overall training objective is formulated as:

$$\mathcal{L}_{\mathrm{total}}(\theta) = (1 - \lambda) \cdot \mathcal{L}_{\mathrm{SFT}}(\theta) + \lambda \cdot \mathcal{L}_{\mathrm{TCD}}(\theta), \tag{7}$$

where $\mathcal{L}_{\mathrm{SFT}}(\theta)$ denotes the autoregressive cross-entropy loss over language outputs, and $\lambda$ is a balancing coefficient for the distillation loss, empirically set to 0.7.

The TCD loss is defined as the Kullback–Leibler (KL) divergence [29, 25] between the output logits distributions of the teacher and student:

$$\mathcal{L}_{\mathrm{TCD}}(\theta) = \mathbb{E}_{\mathcal{I}, \mathcal{P}, t}\big[\mathrm{KL}\big(p^{\mathsf{tea}} \parallel p^{\mathsf{stu}}\big)\big], \tag{8}$$

where $p^{\mathsf{tea}} = \mathrm{Softmax}(\mathbf{h}^{\mathsf{tea}}/\tau)$ and $p^{\mathsf{stu}} = \mathrm{Softmax}(\mathbf{h}^{\mathsf{stu}}/\tau)$ are the temperature-scaled output logits distributions from the teacher and student, respectively, and $\tau$ is a temperature hyperparameter.

By progressively increasing both the student's compression range $\big[R_{\min,t}^{\mathsf{stu}}, R_{\max,t}^{\mathsf{stu}}\big]$ and the teacher–student gap $\Delta_t$, we implement an progressive learning in a token-wise manner. At early training stages, the student experiences mild compression and benefits from a closely aligned teacher, while in later stages, it endures stronger compression with increasingly distinct teacher guidance.

---

[1]Any plug-and-play token compressor (*e.g.*, FastV [7], DART [64]) can serve as compression operator here.

## 3.4 Layer Consistency Distillation

Prior work [7, 76] shows visual tokens receive negligible attention in deeper layers while shallow layers play a more critical role for visual modality. This motivates our **Layer Consistency Distillation (LCD)** strategy: performing compression in deeper layers first minimizes output perturbation and feature space distortion, then progressively moves to shallower layers.

Let $L$ be the total number of Transformer layers in the language model. Define a normalized training progress: $\beta_t = \frac{t}{T}$. We then select a single compression layer for both teacher and student:

$$\ell_t = \text{Round}\big(L - \beta_t(L - \ell_{\min})\big), \quad \ell_t \in \ell_{\min}, \ell_{\min} + 1, \ldots, L \tag{9}$$

where $\ell_{\min}$ denotes the shallowest compression layer. Thus at $t = 0$ we compress at the deepest layer $\ell_0 = L$, and by $t = T$ at the shallowest layer $\ell_T = \ell_{\min}$, realizing a layer-wise progressive learning.

At training iteration $t$, we sample the student model's token compression ratio from $[r_{\min}, r_{\max}]$:

$$r_t^{\text{stu}} \sim \mathcal{U}(r_{\min}, r_{\max}), \tag{10}$$

Typically, $r_{\min}$ is set to 0.2, while $r_{\max}$ is configured as 0.9 and define the teacher model's compression ratio by subtracting the compression ratio gap $\Delta_t$:

$$r_t^{\text{tea}} = \max\big(0, \ r_t^{\text{stu}} - \Delta_t\big). \tag{11}$$

We then perform two forward passes through the shared model $f_\theta$ at layer $\ell_t$:

$$\mathbf{h}^{\text{tea}} = f_\theta\big(C(\mathcal{I}, r_t^{\text{tea}}, \ell_t); \mathcal{P}\big); \quad \mathbf{h}^{\text{stu}} = f_\theta\big(C(\mathcal{I}, r_t^{\text{stu}}, \ell_t); \mathcal{P}\big). \tag{12}$$

The Layer Consistency Distillation (LCD) loss is defined analogously to TCD, as the KL divergence between the teacher and student output logits distributions:

$$\mathcal{L}_{\text{LCD}}(\theta) = \mathbb{E}_{\mathcal{I}, \mathcal{P}, t}\big[\text{KL}(p^{\text{tea}} \| p^{\text{stu}})\big]; \quad p^{\text{tea}} = \text{Softmax}(\mathbf{h}^{\text{tea}}/\tau), p^{\text{stu}} = \text{Softmax}(\mathbf{h}^{\text{stu}}/\tau). \tag{13}$$

Finally, we integrate both distillation terms into the overall training objective:

$$\mathcal{L}_{\text{total}}(\theta) = (1 - \lambda) \cdot \mathcal{L}_{\text{SFT}}(\theta) + \lambda \cdot \mathcal{L}_{\text{LCD}}(\theta). \tag{14}$$

# 4 Experiments

## 4.1 Experimental Setting

**Implementation Details.** We implement EPIC based on LLaVA [41, 40] without introducing any modifications to the model architecture. Specifically, we adopt CLIP ViT-L/14 [49] as our vision encoder, utilizing its officially pretrained projector, and Vicuna-v1.5 [12] as our LLM. algname only requires performing the second stage training, which involves visual instruction tuning on the LLaVA-665K instruction fine-tuning dataset. To demonstrate the effectiveness and generalizability of our method, we incorporate three representative token compression techniques (DART [64], FastV [7], and Random token pruning) into EPIC for training. More implementation details about our proposed method and baselines are provided in Appendix D.

**Evaluation Benchmarks.** We evaluate our model across 10 representative visual understanding benchmarks. Further details about benchmarks can be found in the Appendix D.3.

**Baselines.** As shown in Table 1, we compare our method with various visual token compression techniques, including QT-LLaVA, MQT-LLaVA [26], LLaMA-VID [34], VoCo-LLaMA [70], Token-Packer [33], and LLaVA-Mini [76]. We also list other MLLM's results for comparison, including BLIP-2 [32], InstructBLIP [15], IDEFICS [30], Qwen-VL [3], Qwen-VL-Chat [3], SPHINX [38], mPLUG-Owl2 [68], and vanilla LLaVA [41, 40]. Please refer to Appendix D.4 for more details.

## 4.2 Experimental Results on Benchmarks

Table 1 presents experimental results on 10 representative visual benchmarks. Notably, our method and MQT-LLaVA are among the few approaches enabling flexible control over token compression ($36 \sim 256$ tokens) with a single trained model, adapting efficiently to varying resource constraints.

Table 1: Performance on 10 visual understanding benchmarks. "Res." is resolution, and '#Vision Tokens' is the number of vision tokens. Both training and inference employ DART as the token compression strategy for our methods. Parentheses in Avg.(%) column show diffs vs. LLaVA-v1.5.

| Methods | LLM | Res. | #Vision Tokens | VQA$^{V2}$ | GQA | VizWiz | SQA$^I$ | VQA$^T$ | POPE | MME | MMB | MMB-CN | OCR Bench | Avg. (%) |
|---|---|---|---|---|---|---|---|---|---|---|---|---|---|---|
| BLIP-2 [32] | Vicuna-13B | 224 | 32 | 65.0 | 41.0 | 19.6 | 61.0 | 42.5 | 85.3 | – | – | – | – | – |
| InstructBLIP [16] | Vicuna-7B | 224 | 32 | 66.3 | 49.2 | 34.5 | 60.5 | 50.1 | 83.9 | 1500 | 36.0 | – | 259 | – |
| InstructBLIP [16] | Vicuna-13B | 224 | 32 | 64.2 | 49.5 | 33.4 | 63.1 | – | 84.1 | 1530 | 36.9 | 17.4 | 252 | – |
| IDEFICS-9B [30] | LLaMA-7B | 224 | 64 | 50.9 | 38.4 | 35.5 | – | 25.9 | 75.3 | 1027 | 48.2 | – | 245 | – |
| IDEFICS-80B [30] | LLaMA-65B | 224 | 64 | 60.0 | 45.2 | 36.0 | – | 30.9 | – | 1076 | 54.5 | 29.1 | 277 | – |
| Qwen-VL [3] | Qwen-7B | 448 | 256 | – | 59.3 | 35.2 | 67.1 | 63.8 | – | 1708 | 38.2 | – | 133 | – |
| Qwen-VL-Chat [3] | Qwen-7B | 448 | 256 | – | 57.5 | 38.9 | 68.2 | 61.5 | – | 1891 | 60.6 | – | 267 | – |
| SPHINX [38] | LLaMA-13B | 224 | 289 | 78.1 | 62.6 | 39.9 | 69.3 | 51.6 | 80.7 | – | 66.9 | – | – | – |
| SPHINX-2k [38] | LLaMA-13B | 762 | 2890 | 80.7 | 63.1 | 44.9 | 70.6 | 61.2 | 87.2 | - | 65.9 | – | – | – |
| mPLUG-Owl2 [68] | LLaMA-7B | 448 | 1024 | 79.4 | 56.1 | 54.5 | 68.7 | 54.3 | – | – | 64.5 | – | – | – |
| Video-LLaVA [37] | Vicuna-7B | 224 | 256 | 65.9 | 60.3 | 48.1 | 66.4 | 51.8 | 83.1 | 1542 | 60.6 | 49.3 | 161 | 55.7 |
| LLaVA-v1.5 [41] | Vicuna-7B | 336 | 576 | 72.2 | 61.9 | 52.5 | 68.3 | 58.1 | 85.9 | 1785 | 64.1 | 55.8 | 319 | 61.4 |
| *LMMs with fewer vision tokens* | | | | | | | | | | | | | | |
| Average-Pooling | Vicuna-7B | 336 | 64 | 63.0 | 55.5 | 48.4 | 68.6 | 52.6 | 79.2 | 1579 | 59.6 | 49.5 | 258 | 55.9 (-5.5) |
| MQT-LLaVA [26] | Vicuna-7B | 336 | 2 | 51.4 | 49.6 | 50.0 | 66.1 | 14.8 | 75.4 | 1402 | 40.5 | | 169 | 46.4 (-15) |
| MQT-LLaVA [26] | Vicuna-7B | 336 | 36 | 62.0 | 57.7 | 53.6 | 69.2 | 28.6 | 82.9 | 1777 | 60.5 | 51.6 | 244 | 55.4 (-6.0) |
| MQT-LLaVA [26] | Vicuna-7B | 336 | 64 | 65.6 | 58.7 | 54.3 | 68.4 | 32.5 | 83.1 | 1810 | 61.3 | 53.7 | 260 | 56.8 (-4.6) |
| MQT-LLaVA [26] | Vicuna-7B | 336 | 128 | 66.2 | 59.8 | 54.6 | 69.3 | 35.7 | 84.3 | 1773 | 62.1 | 53.6 | 266 | 57.6 (-3.8) |
| MQT-LLaVA [26] | Vicuna-7B | 336 | 192 | 66.9 | 59.9 | 54.6 | 69.1 | 35.8 | 85.1 | 1784 | 62.0 | 53.9 | 263 | 57.7 (-3.7) |
| MQT-LLaVA [26] | Vicuna-7B | 336 | 256 | 68.3 | 60.1 | 54.6 | 69.0 | 37.1 | 84.6 | 1740 | 61.7 | 53.0 | 273 | 57.8 (-3.6) |
| QT-LLaVA | Vicuna-7B | 336 | 256 | – | 60.3 | 51.5 | 68.1 | 36.9 | 84.1 | 1771 | 62.1 | 53.9 | 265 | – |
| LLaMA-VID [34] | Vicuna-7B | 336 | 2 | – | 55.5 | 54.2 | 68.8 | 49.0 | 83.1 | – | – | – | – | – |
| VoCo-LLaMA [70] | Vicuna-7B | 336 | 1 | – | 55.6 | 54.6 | 68.4 | 31.7 | 80.8 | 1594 | 56.4 | 46.2 | 69 | – |
| TokenPacker [33] | Vicuna-7B | 336 | 144 | 71.3 | 62.0 | 54.6 | 70.5 | 43.8 | 86.2 | 1716 | 63.9 | 53.4 | 303 | 59.9 (-1.5) |
| TokenPacker [33] | Vicuna-7B | 336 | 36 | – | 58.6 | 50.2 | – | – | 83.7 | – | 62.8 | – | – | – |
| LLaVA-Mini [76] | Vicuna-7B | 336 | 144 | 58.1 | 56.3 | 14.8 | 25.3 | 26.0 | 82.3 | 1325 | 24.8 | – | 132 | – |
| LLaVA-Mini [76] | Vicuna-7B | 336 | 64 | - | 56.6 | 10.4 | 27.4 | 28.1 | 82.3 | 1324 | 23.8 | – | 145 | – |
| *Ours* | | | | | | | | | | | | | | |
| LLaVA-v1.5 + TCD | Vicuna-7B | 336 | 256 | 72.7 | 61.4 | 54.1 | 69.8 | 57.0 | 85.8 | 1807 | 66.1 | 54.8 | 310 | 61.7 (+0.3) |
| LLaVA-v1.5 + TCD | Vicuna-7B | 336 | 192 | 71.6 | 60.9 | 54.0 | 70.0 | 56.9 | 85.3 | 1813 | 65.8 | 54.6 | 304 | 61.4 (+0.0) |
| LLaVA-v1.5 + TCD | Vicuna-7B | 336 | 128 | 69.7 | 59.9 | 54.9 | 70.8 | 56.6 | 84.5 | 1861 | 65.6 | 54.3 | 299 | 61.3 (-0.1) |
| LLaVA-v1.5 + TCD | Vicuna-7B | 336 | 64 | 66.1 | 57.1 | 55.1 | 71.1 | 54.8 | 79.2 | 1809 | 64.2 | 53.0 | 286 | 59.4 (-2.0) |
| LLaVA-v1.5 + TCD | Vicuna-7B | 336 | 36 | 62.1 | 54.9 | 55.2 | 71.3 | 53.6 | 75.8 | 1747 | 62.4 | 51.5 | 262 | 57.5 (-3.9) |
| LLaVA-v1.5 + LCD | Vicuna-7B | 336 | 256 | 72.6 | 62.0 | 57.4 | 69.8 | 56.8 | 86.1 | 1834 | 64.3 | 56.0 | 312 | 62.2 (+0.8) |
| LLaVA-v1.5 + LCD | Vicuna-7B | 336 | 192 | 71.3 | 61.5 | 57.6 | 70.0 | 56.7 | 85.3 | 1830 | 64.4 | 55.9 | 316 | 62.0 (+0.6) |
| LLaVA-v1.5 + LCD | Vicuna-7B | 336 | 128 | 69.2 | 60.6 | 57.9 | 69.8 | 56.3 | 84.2 | 1832 | 64.1 | 55.3 | 306 | 61.3 (-0.1) |
| LLaVA-v1.5 + LCD | Vicuna-7B | 336 | 64 | 66.0 | 58.3 | 57.8 | 69.7 | 54.3 | 81.2 | 1794 | 62.1 | 52.9 | 280 | 59.4 (-2.0) |
| LLaVA-v1.5 + LCD | Vicuna-7B | 336 | 36 | 62.8 | 56.5 | 56.5 | 70.3 | 52.9 | 77.7 | 1711 | 60.7 | 51.0 | 265 | 57.6 (-3.8) |

When retaining 128 tokens, our framework achieves performance comparable to vanilla LLaVA-v1.5-7B, while surpassing it with $192+$ visual tokens, strongly suggesting significant redundancy in visual tokens. Compared to other training-aware methods involving model modifications (*e.g.*, MQT-LLaVA, TokenPacker), our approach maintains superior average performance, particularly excelling on MME, MMBench, and VQA V2. This indicates that effective training strategies are as crucial as architectural modifications for token compression. The results also demonstrate our method's robustness across compression ratios: with just 64 tokens, performance degrades by merely 2% versus vanilla LLaVA, while maintaining minimal ($< 1\%$) variation at $128 \sim 256$ visual tokens.

## 4.3 Efficiency

Table 2: Inference efficiency analysis of EPIC. $\Delta$ denotes the reduction ratio. All experiments are on POPE ($8,910$ samples) using an A100 GPU. Token compression is fixed at the 2nd layer.

| Method | Visual Tokens | KV cache (MB) ↓ | | CUDA Time (s) ↓ | | FLOPs (T) ↓ | |
|---|---|---|---|---|---|---|---|
| | | Value | $\Delta$ | Value | $\Delta$ | Value | $\Delta$ |
| LLaVA-v1.5-7B | 576 | 367.2 | – | 1103.5 | – | 9.3 | – |
| EPIC + FastV [7] | 64 | 40.9 | 88.9% | 749.1 | 32.1% | 1.5 | 83.9% |
| EPIC + DART [64] | 64 | 40.9 | 88.9% | 744.3 | 32.6% | 1.5 | 83.9% |
| EPIC + Random | 64 | 40.9 | 88.9% | 697.3 | 36.8% | 1.5 | 83.9% |

We discuss the efficiency of EPIC, including **training-time efficiency** and **inference-time efficiency**. As shown in Table 6 in Appendix D.2, in contrast to other training-aware token compression method, especially those that alter the model architecture, our approach only requires supervised fine-tuning, completing training in approximately 12 hours on 8 A100 GPUs. Most token compression methods that modify the model architecture require two or even three training stages. Furthermore, the replacement or addition of new model components necessitates more training iterations to properly

Table 3: Ablation study on Token Consistency Distillation. "w/o Distillation Loss" disables teacher supervision by zeroing $\mathcal{L}_{\text{TCD}}$. "w/o Progressive Compression Ratio" uses a fixed $88.9\%$ compression ratio. DART is employed as the token compression strategy during training.

| Method | VQA$^{V2}$ | GQA | VizWiz | SQA$^I$ | VQA$^T$ | POPE | MME | MMB | MMB-CN | OCRBench | Avg. (%) |
|---|---|---|---|---|---|---|---|---|---|---|---|
| *Retain 128 Tokens* (↓ **77.8%**) | | | | | | | | | | | |
| **TCD (Ours)** | 69.7 | 59.9 | 54.9 | 70.8 | 56.6 | 84.5 | 1861 | 65.6 | 54.3 | 299 | 61.3 |
| w/o Distillation Loss | 67.2 | 60.4 | 53.2 | 68.6 | 57.3 | 83.2 | 1745 | 63.8 | 53.5 | 289 | 59.8 (-1.5) |
| w/o Progressive Compression Ratio | 67.1 | 58.9 | 49.7 | 70.0 | 54.1 | 84.3 | 1788 | 63.8 | 51.5 | 277 | 59.1 (-2.2) |
| *Retain 64 Tokens* (↓ **88.9%**) | | | | | | | | | | | |
| **TCD (Ours)** | 66.1 | 57.1 | 55.1 | 71.1 | 54.8 | 79.2 | 1809 | 64.2 | 53.0 | 286 | 59.4 |
| w/o Distillation Loss | 65.7 | 57.9 | 53.6 | 69.8 | 54.9 | 78.3 | 1671 | 61.6 | 52.1 | 272 | 58.1 (-1.3) |
| w/o Progressive Compression Ratio | 64.3 | 57.9 | 50.3 | 70.3 | 52.0 | 82.7 | 1774 | 62.8 | 52.5 | 255 | 58.2 (-1.2) |

adapt these parameters, resulting in significantly greater computational expenditure (*e.g.*, $30 \sim 48$ hours on $8$ A100 GPUs). This substantially increases the overall training cost. Focus on the inference-time efficiency, which in our framework is primarily influenced by the token compression strategies employed during the inference process. Table 2 presents the KV cache memory usage, CUDA time, and FLOPs obtained when applying three different token compression methods during inference for models trained using Token Consistency Distillation. The FLOPs are calculated using `calflops` [69], while the KV cache memory usage is estimated with the help of `LLM-Viewer` [72]. It can be observed that when retaining 64 tokens, all methods achieve improvements in reducing KV cache memory, FLOPs, and latency. In particular, Random token compression, which incurs no additional computational overhead, achieves an actual speedup of nearly $1.6\times$.

## 5 Analyses

In this section, we conduct a thorough investigation aimed at addressing the following key questions: (**1**) How much does the weight-sharing teacher guidance contribute to performance? (**2**) What occurs without progressive learning in token/layer-wise dimensions? (**3**) How generalizable is EPIC across various token compression strategies? (**4**) Is extreme token compression (1 or 2 tokens) necessary?

### 5.1 Ablation Studies

Table 4: Ablation study on Layer Consistency Distillation. "w/o Progressive Compression Layer": fixed at the 2nd layer. DART is employed as the token compression strategy during training.

| Method | VQA$^{V2}$ | GQA | VizWiz | SQA$^I$ | VQA$^T$ | POPE | MME | MMB | MMB-CN | OCRBench | Avg. (%) |
|---|---|---|---|---|---|---|---|---|---|---|---|
| *Retain 128 Tokens* (↓ **77.8%**) | | | | | | | | | | | |
| **LCD (Ours)** | 69.2 | 60.6 | 57.9 | 69.8 | 56.3 | 84.2 | 1832 | 64.1 | 55.3 | 306 | 61.3 |
| w/o Distillation Loss | 67.1 | 60.6 | 55.4 | 70.3 | 56.1 | 84.3 | 1761 | 62.9 | 55.4 | 301 | 60.5 (-0.8) |
| w/o Progressive Compression Layer | 68.7 | 59.3 | 54.3 | 70.6 | 56.2 | 82.2 | 1776 | 63.1 | 54.9 | 298 | 60.3 (-1.0) |
| *Retain 64 Tokens* (↓ **88.9%**) | | | | | | | | | | | |
| **LCD (Ours)** | 66.0 | 58.3 | 57.8 | 69.7 | 54.3 | 81.2 | 1794 | 62.1 | 52.9 | 280 | 59.4 |
| w/o Distillation Loss | 64.4 | 58.2 | 55.9 | 69.6 | 54.8 | 80.9 | 1735 | 61.3 | 52.7 | 275 | 58.7 (-0.7) |
| w/o Progressive Compression Layer | 63.5 | 56.5 | 54.7 | 71.5 | 54.2 | 75.6 | 1734 | 61.9 | 51.7 | 260 | 57.8 (-1.6) |

To validate the guiding role of the weight-sharing teacher model (RQ1) and the importance of the progressive learning strategy in training-aware token compression (RQ2), we conducted ablation studies on both components. Specifically, for token consistency distillation (TCD) and layer consistency distillation (LCD), in addition to eliminating the distillation loss to remove the teacher model's influence, we also fix the compression ratio (*e.g.*, $88.9\%$) and compression layer (*e.g.*, the 2nd layer) to eliminate the progressive learning strategy in both the token-wise and layer-wise dimensions. This essentially degenerates the process into imposing a significant perturbation in the feature space, and the model is trained to adapt in the parameter space, like direct training. As demonstrated in Tables 3 and 4, the experimental results indicate that without teacher guidance, both TCD and LCD exhibit significant performance degradation across multiple benchmarks on average, with particularly pronounced declines on vision-centric benchmarks such as MME and MMBench.

## 5.2 How Well Does Proposed Framework Generalize across Different Methods?

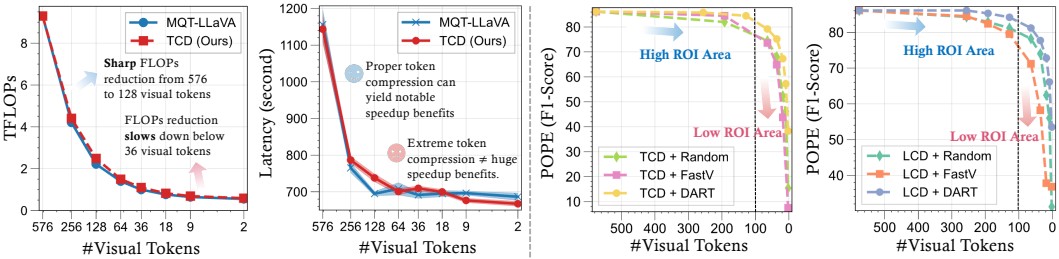

Figure 4: Following LLaVA-v1.5's architecture and data, we apply **DART** for token consistency distillation. "w/o train" denotes vanilla LLaVA. At inference, all methods use $88.9\%$ token compression.

Beyond the generalization across different compression ratios observed in our comparative experiments of Sec. 4.2, we further investigate the adaptability of our proposed framework to diverse token compression strategies (RQ3). Specifically, as outlined in Sec. 4.1, we integrate three plug-and-play token compression strategies into our framework during training. We then conduct cross-strategy evaluations to assess how models trained with one specific strategy (*e.g.*, FastV, DART) perform when applied with alternative compression methods during inference. As shown in Figure 4, token consistency distillation consistently improves model performance across all benchmarks and compression methods. Notably, even when trained solely with DART-based compression, the model generalizes well to FastV and Random compression, yielding consistent performance gains. Furthermore, after training with our proposed framework, the performance gap between token compression strategies is significantly reduced. Notably, previously underperforming strategies exhibit more substantial improvements than their stronger counterparts. For additional experiments and discussions on the generalization capability of our method, please refer to Appendix A.

## 5.3 Is Extreme Token Compression Necessary?

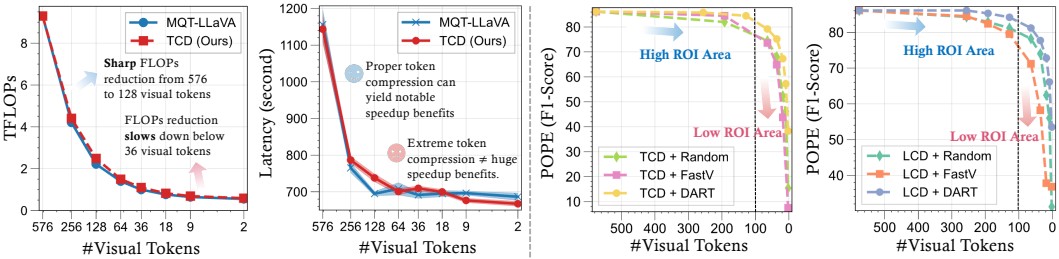

Figure 5: All experiments use the model trained following LLaVA-v1.5. FLOPs and latency are measured on the POPE. Visual token and latency experiments are repeated three times for reliability.

We observe that many token compression methods pursue aggressive compression (*e.g.*, one or two tokens). Table 1 shows that extreme token compression still leads to notable performance degradation for many methods. While this consistently reduces KV cache memory, it raises the question: *does such extreme compression always translate to faster inference* (RQ4)? To answer this, we conducted detailed analysis and experiments. Figure 5 shows the relationship between FLOPs and the number of retained visual tokens. When reducing tokens from the full set (576 tokens) to 128, FLOPs drop significantly—from 9.3T to around 2T. However, under more extreme compression, FLOPs reduction becomes noticeably smaller. A similar or even more pronounced trend is observed between token count and actual latency. In some cases, retaining 64 tokens yields better performance than fewer tokens (*e.g.*, 36 or 18). A possible hypothesis is that overly fragmented feature slices increase memory access time. Moreover, reducing tokens to 64 largely preserves vanilla model performance. We refer to this range as the High Return-on-Investment **(High ROI)** Area. Further reduction beyond 64 offers only marginal latency gains but sharply degrades performance—this is the **Low ROI** Area. Model efficiency depends on whether it is computation or memory-bound. With heavily reduced tokens, GPU compute is underutilized and latency is dominated by memory access, making the

system memory-bound, where further reduction brings little speedup. Overall, we argue that extreme compression is unnecessary; instead, focus should be on balancing latency and performance.

# 6 Conclusion

In this paper, we propose `EPIC`, a learning framework that enhances the efficiency of multi-modal large language models (MLLMs) via Progressive Consistency Distillation. It integrates with existing token compression strategies without modifying the model architecture, achieving efficiency in both training and inference. Experimental results demonstrate that MLLMs trained with our framework achieve comparable average performance to the vanilla model using only 128 visual tokens. Notably, on 4 out of 10 visual understanding benchmarks, our approach even outperforms the vanilla model despite using significantly fewer visual tokens. Furthermore, extensive experiments validate the effectiveness of `EPIC`, highlighting its robustness across token compression ratios and generalization across strategies. Meanwhile, our analysis reveals that while token compression offers significant benefits, excessive compression may lead to a poor latency-performance trade-off, underscoring the importance of balancing compression levels for optimal performance.

# Acknowledgements

This research was supported by the Shanghai Science and Technology Program (Grant No. 25ZR1402278) and Shanghai Artificial Intelligence Laboratory. Besides, we thank Huawei Ascend Cloud Ecological Development Project for the support of Ascend 910 processors.

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

# Appendix

# A  More Generalizability Experiments

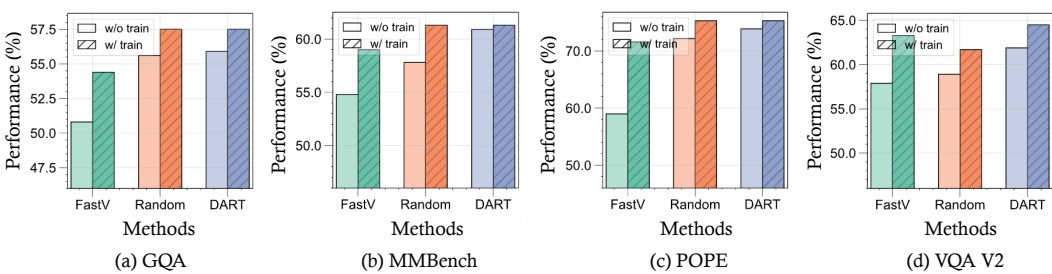

Figure 6: Following model architecture and training data of LLaVA-v1.5-7B, we apply the **Random token compression** for token consistency distillation during training. Here, "w/o train" denotes the vanilla LLaVA. During inference, all methods achieve a token compression ratio of 88.9%.

Figure 7: Following model architecture and training data of LLaVA-v1.5-7B, we apply the **FastV** for token consistency distillation during training. Here, "w/o train" denotes the vanilla LLaVA. During inference, all methods achieve a token compression ratio of 88.9%.

In addition to the cross-method generalization validation of our proposed framework conducted in Sec. 5.2, we performed more comprehensive and meticulous experiments to further verify its

effectiveness. Specifically, as illustrated in Figures 6 and 7, we employ Random Token Compression and FastV, respectively, to perform token consistency distillation on the multi-modal large language model. Experimental results demonstrate that regardless of the token compression strategy used during training with our proposed method, the trained model consistently achieves improved inference performance across various token compression strategies. This strongly demonstrates the generalizability and effectiveness of our approach, indicating that models trained under our framework genuinely adapt to the pattern of missing visual tokens without being constrained to a fixed set of preserved tokens.

# B Detailed Theoretical Analysis

*Proof of Theorem 1.* **1. Direct path.** Since $\theta_{r_t}^{\mathrm{dir}} = c(r_t)$, the total variation is

$$\mathrm{TV}(\{\theta_{r_t}^{\mathrm{dir}}\}) = \sum_{t=0}^{T-1} (c(r_{t+1}) - c(r_t)).$$

By monotonicity **(S1)** and Lipschitz continuity **(S2)**, each term satisfies

$$c(r_{t+1}) - c(r_t) \le \gamma(r_{t+1} - r_t),$$

so summing yields:

$$\mathrm{TV}(\{\theta_{r_t}^{\mathrm{dir}}\}) \le \gamma \sum_{t=0}^{T-1} (r_{t+1} - r_t) = \gamma r_{\max}.$$

**2. Progressive path.** From the closed-form minimizer,

$$\theta_{r_t}^{\mathrm{prog}} = \frac{c(r_t) + \lambda\, c(r_t - \Delta)}{1 + \lambda},$$

we have:

$$\theta_{r_{t+1}}^{\mathrm{prog}} - \theta_{r_t}^{\mathrm{prog}} = \frac{(c(r_{t+1}) - c(r_t)) + \lambda(c(r_{t+1} - \Delta) - c(r_t - \Delta))}{1 + \lambda}.$$

Define $\Delta_t := c(r_{t+1}) - c(r_t) \ge 0$ and $\tilde{\Delta}_t := c(r_{t+1} - \Delta) - c(r_t - \Delta) \ge 0$. Then:

$$\left|\theta_{r_{t+1}}^{\mathrm{prog}} - \theta_{r_t}^{\mathrm{prog}}\right| \le \frac{\Delta_t + \lambda\tilde{\Delta}_t}{1 + \lambda} = \frac{1 + \lambda\kappa_t}{1 + \lambda} \cdot \Delta_t, \quad \text{where } \kappa_t := \frac{\tilde{\Delta}_t}{\Delta_t}.$$

**3. Bounding $\kappa_t$.** Define the worst-case slope ratio:

$$\kappa := \sup_{r \in [\Delta,\, r_{\max} - \Delta]} \frac{c(r) - c(r - \Delta)}{c(r + \Delta) - c(r)}.$$

Each $\kappa_t$ compares the lagged slope to the current slope. If the schedule is appropriately spaced so that $r_t \in [\Delta, r_{\max} - \Delta]$, then we have:

$$\kappa_t = \frac{c(r_{t+1} - \Delta) - c(r_t - \Delta)}{c(r_{t+1}) - c(r_t)} \le \kappa.$$

Convexity **(S3)** ensures that $c'(r)$ is non-decreasing, so

$$c(r) - c(r - \Delta) \le c(r + \Delta) - c(r) \quad \Rightarrow \quad \kappa \le 1.$$

If $c$ is not affine (i.e., $c''(r) > 0$ on a set of positive measure), the inequality is strict for some $r$, and thus $\kappa < 1$.

**4. Total variation bound.** Summing over steps gives:

$$\mathrm{TV}(\{\theta_{r_t}^{\mathrm{prog}}\}) = \sum_{t=0}^{T-1} \left|\theta_{r_{t+1}}^{\mathrm{prog}} - \theta_{r_t}^{\mathrm{prog}}\right| \le \sum_{t=0}^{T-1} \frac{1 + \lambda\kappa}{1 + \lambda} \cdot \Delta_t = \frac{1 + \lambda\kappa}{1 + \lambda} \cdot \mathrm{TV}(\{\theta_{r_t}^{\mathrm{dir}}\}),$$

with strict inequality when $\kappa < 1$, completing the proof. $\qquad\square$

Table 5: Performance on 10 visual understanding benchmarks. 'Res.' is resolution, and '#Vision Tokens' is the number of vision tokens fed to the LLM backbone. Both training and inference of **Integrated Progressive Consistency Distillation (ICD)** employ DART as the token compression strategy for our methods. Parentheses in Avg.(%) column show diffs vs. LLaVA-v1.5.

| Methods | LLM | Res. | #Vision Tokens | VQA$^{V2}$ | GQA | VizWiz | SQA$^I$ | VQA$^T$ | POPE | MME | MMB | MMB-CN | OCR Bench | Avg. (%) |
|---|---|---|---|---|---|---|---|---|---|---|---|---|---|---|
| **BLIP-2** [32] | Vicuna-13B | 224 | 32 | 65.0 | 41.0 | 19.6 | 61.0 | 42.5 | 85.3 | – | – | – | – | – |
| **InstructBLIP** [16] | Vicuna-7B | 224 | 32 | 66.3 | 49.2 | 34.5 | 60.5 | 50.1 | 83.9 | 1500 | 36.0 | – | 259 | |
| **InstructBLIP** [16] | Vicuna-13B | 224 | 32 | 64.2 | 49.5 | 33.4 | 63.1 | – | 84.1 | 1530 | 36.9 | 17.4 | 252 | |
| **IDEFICS-9B** [30] | LLaMA-7B | 224 | 64 | 50.9 | 38.4 | 35.5 | – | 25.9 | 75.3 | 1027 | 48.2 | – | 245 | – |
| **IDEFICS-80B** [30] | LLaMA-65B | 224 | 64 | 60.0 | 45.2 | 36.0 | – | 30.9 | – | 1076 | 54.5 | 29.1 | 277 | – |
| **Qwen-VL** [3] | Qwen-7B | 448 | 256 | – | 59.3 | 35.2 | 67.1 | 63.8 | – | 1708 | 38.2 | – | 133 | – |
| **Qwen-VL-Chat** [3] | Qwen-7B | 448 | 256 | – | 57.5 | 38.9 | 68.2 | 61.5 | – | 1891 | 60.6 | – | 267 | – |
| **SPHINX** [38] | LLaMA-13B | 224 | 289 | 78.1 | 62.6 | 39.9 | 69.3 | 51.6 | 80.7 | – | 66.9 | – | – | – |
| **SPHINX-2k** [38] | LLaMA-13B | 762 | 2890 | 80.7 | 63.1 | 44.9 | 70.6 | 61.2 | 87.2 | – | 65.9 | – | – | – |
| **mPLUG-Owl2** [68] | LLaMA-7B | 448 | 1024 | 79.4 | 56.1 | 54.5 | 68.7 | 54.3 | - | – | 64.5 | – | – | – |
| **Video-LLaVA** [37] | Vicuna-7B | 224 | 256 | 65.9 | 60.3 | 48.1 | 66.4 | 51.8 | 83.1 | 1542 | 60.6 | 49.3 | 161 | 55.7 |
| **LLaVA-v1.5** [41] | Vicuna-7B | 336 | 576 | 72.2 | 61.9 | 52.5 | 68.3 | 58.1 | 85.9 | 1785 | 64.1 | 55.8 | 319 | 61.4 |
| *LMMs with fewer vision tokens* | | | | | | | | | | | | | | |
| **Average-Pooling** | Vicuna-7B | 336 | 64 | 63.0 | 55.5 | 48.4 | 68.6 | 52.6 | 79.2 | 1579 | 59.6 | 49.5 | 258 | 55.9 (-5.5) |
| **MQT-LLaVA** [26] | Vicuna-7B | 336 | 2 | 51.4 | 49.6 | 50.0 | 66.1 | 14.8 | 75.4 | 1402 | 48.9 | 40.5 | 169 | 46.4 (-15) |
| **MQT-LLaVA** [26] | Vicuna-7B | 336 | 36 | 62.0 | 57.7 | 53.6 | 69.2 | 28.6 | 82.9 | 1777 | 60.5 | 51.6 | 244 | 55.4 (-6.0) |
| **MQT-LLaVA** [26] | Vicuna-7B | 336 | 64 | 65.6 | 58.7 | 54.3 | 68.4 | 32.5 | 83.1 | 1810 | 61.3 | 53.7 | 260 | 56.8 (-4.6) |
| **MQT-LLaVA** [26] | Vicuna-7B | 336 | 128 | 66.2 | 59.8 | 54.6 | 69.3 | 35.7 | 84.3 | 1773 | 62.1 | 53.6 | 266 | 57.6 (-3.8) |
| **MQT-LLaVA** [26] | Vicuna-7B | 336 | 192 | 66.9 | 59.9 | 54.6 | 69.1 | 35.8 | 85.1 | 1784 | 62.0 | 53.9 | 263 | 57.7 (-3.7) |
| **MQT-LLaVA** [26] | Vicuna-7B | 336 | 256 | 68.3 | 60.1 | 54.6 | 69.0 | 37.1 | 84.6 | 1740 | 61.7 | 53.0 | 273 | 57.8 (-3.6) |
| **QT-LLaVA** | Vicuna-7B | 336 | 256 | – | 60.3 | 51.5 | 68.1 | 36.9 | 84.1 | 1771 | 62.1 | 53.9 | 265 | – |
| **LLaMA-VID** [34] | Vicuna-7B | 336 | 2 | – | 55.5 | 54.2 | 68.8 | 49.0 | 83.1 | – | – | – | – | – |
| **VoCo-LLaMA** [70] | Vicuna-7B | 336 | 1 | – | 55.6 | 54.6 | 68.4 | 31.7 | 80.8 | 1594 | 56.4 | 46.2 | 69 | – |
| **TokenPacker** [33] | Vicuna-7B | 336 | 144 | 71.3 | 62.0 | 56.6 | 70.5 | 43.8 | 86.2 | 1716 | 63.9 | 53.4 | 303 | 59.9 (-1.5) |
| **TokenPacker** [33] | Vicuna-7B | 336 | 36 | – | 58.6 | 50.2 | – | – | 83.7 | – | 62.8 | – | – | – |
| **LLaVA-Mini** [76] | Vicuna-7B | 336 | 144 | 58.1 | 56.3 | 14.8 | 25.3 | 26.0 | 82.3 | 1325 | 24.8 | – | 132 | – |
| **LLaVA-Mini** [76] | Vicuna-7B | 336 | 64 | - | 56.6 | 10.4 | 27.4 | 28.1 | 82.3 | 1324 | 23.8 | – | 145 | – |
| *Ours* | | | | | | | | | | | | | | |
| **LLaVA-v1.5 + ICD** | Vicuna-7B | 336 | 256 | 72.2 | 61.7 | 54.7 | 70.0 | 58.1 | 85.7 | 1788 | 64.9 | 55.7 | 312 | 61.8 (+0.4) |
| **LLaVA-v1.5 + ICD** | Vicuna-7B | 336 | 192 | 71.9 | 61.4 | 54.6 | 70.3 | 57.6 | 84.8 | 1783 | 64.8 | 55.9 | 316 | 61.7 (+0.3) |
| **LLaVA-v1.5 + ICD** | Vicuna-7B | 336 | 128 | 69.6 | 60.3 | 55.2 | 70.1 | 57.1 | 83.5 | 1770 | 63.9 | 55.2 | 298 | 60.8 (-0.6) |
| **LLaVA-v1.5 + ICD** | Vicuna-7B | 336 | 64 | 66.5 | 57.5 | 55.3 | 70.4 | 54.7 | 79.3 | 1720 | 63.8 | 53.2 | 275 | 59.0 (-2.4) |
| **LLaVA-v1.5 + ICD** | Vicuna-7B | 336 | 36 | 62.1 | 55.6 | 54.4 | 70.9 | 52.8 | 74.0 | 1635 | 61.2 | 51.1 | 246 | 56.5 (-4.9) |

# C  Integrated Progressive Consistency Distillation

As outlined in Sec. 1, our Progressive Consistency Distillation framework incorporates two progressive learning mechanisms: Token Consistency Distillation (TCD) in a token-wise manner and Layer Consistency Distillation (LCD) in a layer-wise manner. Experimental results demonstrate that both approaches achieve strong performance across all key metrics: model accuracy, efficiency, robustness, and generalization capability. To further investigate this direction, we explore the integration of progressive learning from both token-wise and layer-wise perspectives. A critical question arises: *Can this integrated approach continue to maintain or even enhance the model's performance?* Building upon these two approaches, we design Integrated Progressive Consistency Distillation (ICD). Unlike the original Token Consistency Distillation (TCD), where token-wise progression is governed by global training progress, our method enables layer-wise iterative application of TCD, effectively integrating both token-wise and layer-wise dimensions. Specifically, during training, the token compression layer progressively shifts from deeper to shallower layers. Within each layer, the compression ratio follows the TCD schedule, sampling from small to large values. Upon transitioning to the next layer, the ratio resets to its initial value, and the process repeats. As shown in Table 5, the Integrated Progressive Consistency Distillation approach achieves comparable performance on representative visual benchmarks. Notably, it even surpasses the vanilla LLaVA in average performance while retaining only 192 visual tokens (66.7% ↓). Furthermore, when compared to other training-aware token compression approaches (such as MQT-LLaVA and TokenPacker), ICD demonstrates superior performance under identical retained visual tokens.

# D    Experimental Setup

## D.1    Token Compression Techniques During Training

We adopt three representative token compression methods in our proposed framework: FastV, which follows an importance-based strategy; DART, which leverages redundancy-based pruning; and random token pruning, the simplest form of token compression.

- DART [64] is a training-free and plug-and-play method that prunes visual tokens based on token duplication while maintaining compatibility with efficient attention mechanisms like Flash Attention [18, 17].

- FastV [7] is a plug-and-play token compression technique that builds on the observation that visual tokens tend to have diminishing contributions to model outputs in deeper layers. By utilizing attention scores to assess token importance, it prunes less critical visual tokens at earlier layers of the language model.

- Random is a token compression method that requires no additional signals or computation, and is primarily used to validate the effectiveness and generalizability of our proposed approach.

Table 6: Method Comparison Across Different Stages

| Method | Stage 1 | Stage 2 | Stage 3 | Training Time ↓ (h) |
|---|---|---|---|---|
| QT-LLaVA | ✓ | ✓ | | $\sim$ 30h $\times$ 8 A100s |
| MQT-LLaVA | ✓ | ✓ | | $\sim$ 34h $\times$ 8 A6000s |
| LLaMA-VID | ✓ | ✓ | ✓ | $\sim$ 48h $\times$ 8 A100s |
| LLaVA-Mini | ✓ | ✓ | | $\sim$ 26h $\times$ 8 A100s |
| EPIC (Ours) | | ✓ | | $\sim$ 12.2h $\times$ 8 A100s |

## D.2    Training Details

For a fair comparison, our framework not only adheres to the same model architecture and instruction-tuning data as vanilla LLaVA but also maintains identical hyperparameter settings. Moreover, since we have not modified the model architecture, we can directly use the pre-trained projector—unlike MQT-LLaVA, QT-LLaVA, and LLaVA-Mini, which require mandatory Stage 1 pre-training. The training process, conducted on 8 $\times$ A100 GPUs, takes approximately 12 hours. Furthermore, we faithfully reproduced the entire LLaVA training process following the official LLaVA-v1.5 training guidelines. All other token compression baselines are either trained following the settings provided in the original papers or evaluated using publicly available model checkpoints. For the ablation study in Section 5.1, the experiment without the distillation loss was trained using the same LLaVA-665K SFT data, with all other training parameters and procedures kept identical to those of EPIC. For the variants without a progressive compression ratio (Table 3) or progressive compression layer (Table 4), we fixed the second layer as the compression layer. Overall, our ablation studies were conducted under a strictly fair and consistent experimental setup.

Table 7: Detailed hyperparameter settings.

| Settings | Stage 2 |
|---|---|
| Batch size | 128 |
| Learning rate | 2e-5 |
| Learning schedule | Cosine decay |
| Warmup ratio | 0.03 |
| Weight decay | 0 |
| Epoch | 1 |
| Optimizer | AdamW |
| DeepSpeed stage | 3 |
| Max token | 2048 |

## D.3 Benchmarks

- MME [21] is a comprehensive benchmark for evaluating the performance of MLLMs in multi-modal tasks. It measures models' capabilities across two key areas: perception and cognition, using 14 specially designed subtasks that test interpretative and analytical skills.

- MMBench [45] employs a dual approach: it provides an extensive dataset that broadens the range and variety of evaluation questions, and introduces the innovative CircularEval strategy, which uses ChatGPT to convert free-form predictions into structured choices. MMBench-CN is the Chinese version of the benchmark.

- ScienceQA [47] is a multi-modal benchmark aimed at assessing and diagnosing AI systems' multi-hop reasoning and interpretability in the science domain. It includes a dataset of around 21K multiple-choice questions across various scientific topics, complete with detailed answer annotations, related lectures, and explanations.

- GQA [27] is a dataset designed for advanced visual reasoning in real-world scenarios, using scene graph-based structures to generate 22 million diverse, semantically-programmed questions. It features a novel set of evaluation metrics focused on consistency, grounding, and plausibility, setting a high standard for vision-language task assessment.

- POPE [36] is an evaluation method for examining object hallucination in MLLMs. It transforms the evaluation into a binary classification task, asking MLLMs simple Yes-or-No questions to identify hallucinated objects. POPE employs various object sampling strategies to reveal model tendencies towards hallucination.

- VQA V2 [22] evaluates the model's visual perception capabilities through open-ended questions. It consists of 265,016 images, covering a wide variety of real-world scenes and objects, providing rich visual contexts for the questions. For each question, there are 10 ground truth answers provided by human annotators, which allows for a comprehensive evaluation of the performance of different models in answering the questions accurately.

- TextVQA [51] focuses on the comprehensive integration of diverse text information within images. It meticulously evaluates the model's text understanding and reasoning abilities through a series of visual question-answering tasks with rich textual information. Models need to not only understand the visual content of the images but also be able to read and reason about the text within the images to answer the questions accurately.

- OCRBench [46] is a comprehensive benchmark for evaluating the OCR capabilities of multi-modal language models across five key tasks: text recognition, scene text-centric and document-oriented VQA, key information extraction, and handwritten mathematical expression recognition.

## D.4 Overview of the Baselines

### D.4.1 General MLLMs

- BLIP-2 [32] is a vision-language pretraining framework that efficiently combines frozen image encoders with large language models (LLMs). It adopts a two-stage training strategy leveraging a lightweight Querying Transformer to bridge the vision-language modality gap, enabling compute-efficient, zero-shot image-to-text generation aligned with natural language instructions.

- InstructBLIP [15] builds on BLIP-2 by introducing instruction tuning and an instruction-aware Query Transformer. This enhances the model's ability to extract features for a wide range of vision-language tasks. It achieved state-of-the-art zero-shot performance across 13 benchmarks and demonstrated strong results on fine-tuned tasks such as ScienceQA.

- IDEFICS [30] is an open-access vision-language model based on the Flamingo [1] architecture. Available in both base and instruction-tuned variants (9B and 80B parameters), IDEFICS is trained entirely on publicly available data and models, promoting transparency and accessibility.

- Qwen-VL & Qwen-VL-Chat [3] extend the Qwen-LM [2] foundation with a visual encoder and a specialized input-output interface. Through a three-stage training pipeline and a rich multilingual, multimodal corpus, the models achieve strong capabilities in grounding and optical character recognition (OCR) tasks.

- SPHINX [38] is a multimodal large language model that applies joint mixing across model weights, tuning objectives, visual embeddings, and image scales. By unfreezing the LLM during pretraining

and integrating diverse instructional and visual signals, SPHINX demonstrates strong performance in fine-grained vision-language understanding and reasoning tasks.

- mPLUG-Owl2 [68] features a modular architecture with a language decoder interface for unified modality coordination. It employs shared cross-modal modules alongside modality-adaptive components to enhance feature retention and generalization in both unimodal and multimodal settings.

- Video-LLaVA [37] extends multimodal language modeling to unified video and image understanding. By aligning visual modalities into a shared language feature space prior to projection, the model enables effective joint training across visual domains, achieving state-of-the-art results on diverse video and image benchmarks without requiring paired video-image data.

- LLaVA-v1.5 [39] improves upon the original LLaVA [41] model through enhanced visual instruction tuning. Utilizing a CLIP-ViT-L-336px visual backbone and MLP-based projection, it achieves strong performance with high data efficiency. Trained on only 1.2 million publicly available images, it excels at academic-oriented VQA tasks using straightforward prompting strategies.

### D.4.2   Training-aware Token Compression Methods

- Average Pooling, inspired by the token merging strategy in Qwen2-VL [58], merges every four adjacent visual patches into a single token, effectively reducing the number of visual tokens.

- QT-LLaVA replaces the original modality projector in LLaVA with a Q-former [32] that uses learnable queries to project the input token sequence into a shorter sequence. In this work, we fix to retain $256$ visual tokens.

- MQT-LLaVA [26] proposes a flexible Query Transformer that enables encoding images into a variable number of visual tokens (up to a predefined maximum), allowing dynamic adaptation to different tasks and computational budgets.

- LLaMA-VID [34] compresses both the instruction and the image into a single token each, resulting in just two tokens per image. This design enables efficient understanding of longer video sequences.

- VoCo-LLaMA [70] utilizes language models to compress all vision tokens, significantly improving computational efficiency while maintaining multimodal understanding.

- TokenPacker [33] introduces a visual projector that adopts a coarse-to-fine strategy to reduce the number of visual tokens by up to $80\%$, substantially decreasing the computational cost.

- LLaVA-Mini [76] employs a modality pre-fusion module, constructed with Transformer blocks, to integrate visual information into the text tokens in advance. This approach reduces the number of visual tokens fed into the language model, thereby lowering computational cost.

## E   Limitations and Future Works

As discussed in the Appendix D.2, our proposed progressive consistency distillation framework was only applied during the visual instruction tuning phase when training the multi-modal large language models (MLLMs). While this approach has already demonstrated remarkable effectiveness in terms of both model performance and training efficiency, several promising research directions remain unexplored. For instance, how might our method perform if applied during the model's pretraining stage? Specifically, implementing our framework during projector pretraining, rather than initializing it with publicly available projector weights, could potentially yield greater performance improvements. We hypothesize that this would allow the modality projector's parameter space to adapt to token compression-induced feature space perturbations prior to fine-tuning, thereby further facilitating subsequent supervised fine-tuning. In future work, we will systematically investigate extending our framework to all training stages (including pretraining and supervised finetuning), with rigorous analysis of both the performance gains and impacts on training-time efficiency.

## F   Broader Impacts

In this work, we propose a framework to develop efficient multi-modal large language models (MLLMs) through Progressive Consistency Distillation, which can be seamlessly integrated with various token compression strategies without modifying the original model architecture. Our approach

demonstrates strong effectiveness, robustness, and generalizability. On one hand, this contributes to improving the efficiency of MLLMs, thereby facilitating their deployment and practical applications at a societal level, especially for resource-constrained edge devices. On the other hand, after training under our framework with token-compressed inputs, the resulting model parameters shift from the optimum of the vanilla model (trained on full inputs) to a new optimum suited for compressed inputs. However, we do not further align these models with human preferences after token compression-based training. While the models perform well across a range of multi-modal tasks, they may carry potential risks of adversarial vulnerabilities or undesirable outputs.

