# OpenReview forum: "Efficient Multi-modal Large Language Models via Progressive Consistency Distillation"
_NeurIPS.cc/2025/Conference — NeurIPS 2025 poster_

### Official Review · Reviewer_Guzs · 2025-06-26

**Clarity:** 3
**Significance:** 2
**Originality:** 3
**Rating:** 4
**Confidence:** 4

**Summary:**

This paper proposes EPIC, a framework that addresses excessive computational resource consumption by visual tokens in multi-modal large language models. The method decomposes token compression perturbations into token-wise and layer-wise dimensions, introducing Token Consistency Distillation (TCD) and Layer Consistency Distillation (LCD) with progressive learning trajectories. Extensive experiments show efficiency improvements.

**Questions:**

Refer to weak points.

**Ethical Concerns:**

["NO or VERY MINOR ethics concerns only"]

**Final Justification:**

I would be maintain my score.

**Limitations:**

Refer to weak points.

**Quality:**

3

**Strengths And Weaknesses:**

Strong points:
1.The progressive consistency distillation approach proposed in this paper demonstrates certain innovation, with progressive learning from both token and layer dimensions, which is inspiring.
2.To prove the effectiveness of the proposed method, this paper conducts a series of comparative and ablation experiments, along with efficiency analysis of the model. The experimental design is reasonably.
3.The proposed method achieves significant improvements in computational efficiency for multi-modal large models: using only 64 tokens achieves merely 2% performance degradation compared to the original model, while substantially improving inference efficiency.

Weak points:
1.The visualization of learning trajectories shown in Figure 1 is rather abstract and difficult to illustrate the problem emphasized in this paper. Quantitative experiments are needed for detailed explanation.
2.To my knowledge, both Token Consistency Distillation (TCD) and Layer Consistency Distillation (LCD) operate on tokens, just from different perspectives. What are the essential differences between the two approaches? A detailed analysis is needed.
3.Layer Consistency Distillation (LCD) follows an easy-to-hard learning paradigm. What are the differences and connections with commonly used curriculum learning methods? There is a lack of deeper analysis on why progressive learning is effective.
4.How are the hyperparameter λ in Section 3.2 selected and determined? The selection lacks sensitivity analysis.
5.This paper demonstrates the effectiveness of the proposed method on visual understanding benchmarks, but the task formats are relatively limited. Additionally, experiments are only conducted using LLaVA, lacking validation of the method's generalizability on other mainstream MLLM architectures (such as BLIP-2[1], InstructBLIP[2], etc.).
[1]Li J, Li D, Savarese S, et al. Blip-2: Bootstrapping language-image pre-training with frozen image encoders and large language models[C]//International conference on machine learning. PMLR, 2023: 19730-19742.
[2]Dai W L, Li J N, Li D X, et al. InstructBLIP: Towards general-purpose vision-language models with instruction tuning[C]//37th Conference on Neural Information Processing Systems (NeurIPS 2023). 2023-05-11.

---

> ### Author Rebuttal · Authors · 2025-07-31
>
> Dear Reviewer Guzs,
>
> Thank you for your insightful feedback. We appreciate your thoughtful comments and are pleased to address each of your concerns in detail below.
>
> > **W1: Quantitative experiments are needed for detailed explanation on Fig.1.**
>
> **RW1:** Thank you for your feedback. While Fig.1 qualitatively illustrates the difficulty of directly training under high compression ratios, we fully agree that quantitative evidence is important. To address this, we conduct controlled experiments comparing direct training and progressive learning under the same training budget (4000 steps) on LLaVA-665K. For direct training, we randomly drop 60% of tokens and fine-tune the pretrained LLaVA-v1.5-7B for 4000 steps. For progressive training, we divide training into four stages (0%, 20%, 40%, 60% pruning), each lasting 1000 steps. In each stage, the student adopts the current pruning ratio, while the teacher (shared weights) retains the previous stage’s pruning ratio to provide a consistency signal. Both methods use random pruning to isolate the effect of progressive learning. **Experimental results in Table below show that progressive distillation significantly outperforms direct training, supporting the motivation of our method.**
>
> |                      Model                      |  MME  |  GQA  | TextVQA |  SQA  | MMbench_en | vizwiz | Average |
> | :---------------------------------------------: | :---: | :---: | :-----: | :---: | :--------: | :----: | :-----: |
> |    LLaVA-v1.5-7B (Random Pruning Ratio 0.6)     | 1700  | 58.5  |  53.8   | 66.7  |    60.7    |  52.8  |  58.9   |
> | Progressive Learning (Random Pruning Ratio 0.6) | 1763  | 60.2  |  55.4   | 68.8  |    62.9    |  54.7  |  60.8   |
>
> We hope this helps better illustrate our motivation, and we will include these experimental results in the final manuscript.
>
> > **W2: A detailed comparison of TCD and LCD**
>
> Thank you for the insightful question. TCD and LCD are two complementary designs within our unified EPIC framework: TCD effects a temporal progression by gradually elevating the compression ratio, whereas LCD embodies a spatial progression by shifting the compression layer from deep to shallow, inspired by the observation that visual tokens receive diminishing attention in deeper layers. We will enrich the revised manuscript with further comparative details to highlight their distinct yet synergistic roles.
>
> > **W3: Differences and connections with curriculum learning and deeper analysis on effectiveness of progressive learning**
>
> Thank you for the insightful comment. We appreciate the opportunity to clarify the conceptual differences and underlying mechanisms.
>
> **Connections and Differences with Curriculum Learning.** We agree that Layer Consistency Distillation (LCD) aligns with the easy-to-hard philosophy inherent to curriculum learning (CL). Both frameworks aim to reduce training difficulty and enhance convergence by gradually increasing task complexity. However, LCD diverges from standard CL in the following ways:
> - **Different Learning Objects:** Traditional CL operates on input samples or labels, selecting or ordering training data based on their estimated difficulty. In contrast, LCD progressively shifts the *token compression layer* from deep to shallow Transformer layers, which affects the model’s internal feature space rather than external data complexity.
> - **Mechanistic Basis:** LCD is grounded in empirical observations (e.g., [1]) that visual tokens receive minimal attention in deeper layers, meaning early compression in these layers causes less disruption. The progressive schedule is thus designed to gradually introduce stronger perturbations in a *structurally informed* manner, rather than relying on heuristic difficulty measures.
> - **Distinct Objective:** While CL generally aims to improve generalization or training efficiency, LCD is specifically tailored to *mitigate the feature-space perturbations* caused by token compression. It allows the model to transition smoothly from the uncompressed optimum to the compressed one, which we find critical for performance under high compression.
>
> **Deeper Analysis on the Effectiveness of Progressive Learning.**
> In Section 3.2, we provide a theoretical prototype demonstrating the benefits of progressive consistency distillation in a simplified 1D setting. We model the shift in optimal outputs (centers) induced by increasing compression ratios, and show that incorporating a teacher with a slightly lower compression ratio leads to a smoother optimization path with *lower total variation*. This translates into better training stability and convergence.
>
> Empirically, we support this with ablations in Section 5.1 (Table 4), where removing progressive layer scheduling leads to performance drops up to 1.6% under heavy compression (64 tokens). This highlights that progressive learning is not only theoretically grounded but also essential in practice.
>
> We hope this addresses your concerns, and we are happy to clarify further.
>
> > **W4: The sensitivity analysis about hyperparameter λ.**
>
> **RW4:** Thank you for raising this insightful question. As mentioned in the main text (line 178), TCD and LCD default to λ = 0.7. Following your suggestion, we conducted a sensitivity analysis on λ by evaluating token consistency distillation with λ ∈ {0.1, 0.3, 0.5, 0.7, 0.9}. As shown in the table below (evaluations performed at 77.8% pruning ratio with DART), we observe that model performance degrades significantly when λ is either too large or too small. When λ approaches 0, the distillation loss is almost disabled, depriving the model of valuable guidance from the teacher. Conversely, when λ approaches 1, the SFT loss is suppressed, which can impair the model's autoregressive generation capability. **Our experiments demonstrate that λ in [0.5, 0.7] yields the most favorable results, which balances the KL guidance from the teacher with the task-specific supervision from SFT.**
>
> | Hyperparameters λ |  MME  | POPE  |  GQA  | TextVQA | MMbench_en | VizWiz | Average |
> | :---------------: | :---: | :---: | :---: | :-----: | :--------: | :----: | :-----: |
> |        0.1        | 1749  | 83.2  | 59.8  |  57.2   |    63.8    |  53.2  |  63.3   |
> |        0.3        | 1845  | 83.8  | 59.4  |  56.4   |    65.3    |  54.6  |  64.2   |
> |        0.5        | 1829  | 84.1  | 60.6  |  57.3   |    66.4    |  55.6  |  64.9   |
> |        0.7        | 1861  | 84.5  | 59.9  |  56.6   |    65.6    |  54.9  |  64.7   |
> |        0.9        | 1770  | 81.8  | 58.2  |  53.9   |    63.1    |  54.1  |  62.4   |
>
> We will incorporate these insights into the revised version of our manuscript. Once again, we sincerely thank you for your valuable feedback, which has significantly strengthened the quality and clarity of our work.
>
> > **W5: Add validation on other MLLM architectures**
>
> **RW5:** Thank you for the insightful suggestion. We have evaluated our method on a range of MLLMs, including LLaVA-Next-7B (featuring a different architectural design), and the Qwen-VL series (distinct architecture). All models are trained on the same LLaVA-665K, under identical training configurations. **As shown in the table below, our approach maintains over 95% of the vanilla model's average performance even when 77.8% of visual tokens are pruned, and achieves nearly full performance at a 66.7% pruning ratio, surpassing direct SFT by up to 4.9% under 66.7% pruning.** These results clearly demonstrate the effectiveness and generalization capability of EPIC across diverse model architectures and scales. We will incorporate these findings into the revised manuscript.
>
> |                   Model                   |  MME  | POPE  |  GQA  | TextVQA |  SQA  | MMbench_en | OCRBench | VizWiz | VQA_v2 |    Average    |
> | :---------------------------------------: | :---: | :---: | :---: | :-----: | :---: | :--------: | :------: | :----: | :----: | :-----------: |
> |               Qwen2.5-VL-3B               | 1738  | 86.6  | 62.5  |  58.4   | 74.2  |    66.8    |   315    |  48.7  |  72.0  |     62.5     |
> | + Direct SFT (Pruning Ratio 0.667) | 1702 | 82.3 | 60.1 | 55.8 | 73.1 | 64.8 | 300 | 48.1 | 69.3 | 60.5 (96.8%) |
> | + TCD (Pruning Ratio 0.667) | 1732  | 85.1  | 62.0  |  57.1   | 74.1  |    66.8    |   301    |  48.3  |  71.3  | 61.9 (99.0%) |
> | + TCD (Pruning Ratio 0.778) | 1711  | 83.7  | 61.4  |  55.6   | 73.5  |    65.3    |   287    |  47.6  |  70.1  | 60.8 (96.9%) |
> ||
> |                Qwen2-VL-7B                | 2212  | 85.9  | 67.3  |  80.1   | 82.1  |    79.2    |   664    |  53.6  |  77.9  |     74.6      |
> | + Direct SFT (Pruning Ratio 0.667) | 2110 | 82.8 | 66.1 | 77.3 | 80.1 | 78.5 | 649 | 52.0 | 75.9 | 72.6 (97.3%) |
> | + TCD (Pruning Ratio 0.667) | 2199  | 84.2  | 67.3  |  78.9   | 82.3  |    80.1    |   657    |  53.7  |  78.0  | 74.3 (99.6%)  |
> | + TCD (Pruning Ratio 0.778) | 2157  | 83.8  | 65.6  |  78.1   | 81.6  |    78.7    |   647    |  52.3  |  76.5  | 73.1 (98.0%)  |
> ||
> |               LLaVA-Next-7B               | 1783  | 87.1  | 63.6  |  67.5   | 68.7  |    67.6    |   540    |  63.1  |  74.7  |     67.8      |
> | + Direct SFT (Pruning Ratio 0.667) | 1715 | 84.2 | 61.4 | 66.0 | 67.3 | 65.3 | 518 | 61.1 | 73.2 | 65.7 (96.9%) |
> | + TCD (Pruning Ratio 0.667) | 1809  | 85.9  | 63.1  |  66.9   | 68.7  |    68.7    |   537    |  62.7  |  73.4  | 67.5 (99.6%)  |
> | + TCD (Pruning Ratio 0.778) | 1740  | 85.4  | 62.2  |  65.1   | 68.3  |    67.9    |   523    |  60.9  |  74.4  | 66.5 (98.1%)  |
>
> **(Due to space limits, more results on LLaVA-v1.5-13B and Qwen2-VL-0.5B can be found in RW1 to Reviewer aYx1, or we can share the results in the discussion thread upon request. Thanks for understanding)**
>
> **Reference**
>
> [1] Chen, Liang, et al. "An image is worth 1/2 tokens after layer 2: Plug-and-play inference acceleration for large vision-language models." European Conference on Computer Vision. Cham: Springer Nature Switzerland, 2024.
>
> Best Regards,
>
> The Authors

---

> > ### Comment · Reviewer_Guzs · 2025-08-06
> >
> > Thank you for your comprehensive and thoughtful response to my feedback. I appreciate the thoroughness with which you have addressed each of my concerns. I am uncertain whether adding new experiments during the rebuttal phase is permitted. Nevertheless, it is very likely that I will maintain my positive score.

---

> ### Author Response · Authors · 2025-08-06
>
> Dear Reviewer Guzs,
>
> We are delighted to hear that our responses have addressed your concerns. Thank you very much for your positive feedback, and for the time and effort you devoted to reviewing our paper and helping us improve it.
>
> Regarding your note on new experiments during the rebuttal phase: we have carefully reviewed the official NeurIPS guidelines, including both the conference website and the related emails, and we confirm that adding new experiments to address reviewers’ concerns is indeed permitted. Accordingly, we have included additional results in our rebuttal to better support and strengthen our claims in response to your valuable feedback.
>
> We truly appreciate your thoughtful comments and will incorporate your suggestions into the final version. Thank you again for your valuable support!
>
> Best regards,
>
> The Authors

---

### Official Review · Reviewer_BQCk · 2025-06-30

**Clarity:** 2
**Significance:** 2
**Originality:** 2
**Rating:** 3
**Confidence:** 4

**Summary:**

This paper focuses on the challenge that visual tokens in multimodal large language models (MLLMs) consume significant computational resources, presenting Efficient MLLMs via Progressive Consistency Distillation (EPIC), a novel progressive learning framework. The proposed method decomposes feature space perturbations induced by token compression along token-wise and layer-wise dimensions, introducing token consistency distillation and layer consistency distillation respectively. By leveraging guidance from a teacher model and following a progressive learning trajectory, EPIC aims to mitigate training difficulties. Experimental results across diverse datasets validate the approach’s effectiveness, demonstrating its capability to achieve high efficiency, robustness, and generalization performance.

**Questions:**

1. For the ablation study section: the direct training without distillation loss, does it use the same dataset and the same training approach? Regarding the configuration without progressive compression, which layers are set? We understand that the selection of layers will affect the effect after compression. Have appropriate and consistent layers been set to maintain the validity of the ablation experiments?
2. Would it be possible to provide more comparative experimental results on different models across varying parameter scales?
3. I noticed that in the experimental section, some compressed models unexpectedly outperform the original models. What could be the reasons behind this phenomenon?
4. Why isn't the result of the complete EPIC method combining TCD and LCD included in the main Table 1 of the paper? Is it possible that the combined effect of the two methods is not ideal?
5. For other vision-language (VL) models, is this method EIPC equally effective?

**Ethical Concerns:**

["NO or VERY MINOR ethics concerns only"]

**Final Justification:**

I still hold my initial view on the novelty of the paper.
While I will not oppose acceptance if the other reviewers are in favor, I believe the motivation and justifications could benefit from further clarification and refinement.

**Limitations:**

yes

**Quality:**

2

**Strengths And Weaknesses:**

Strengths:
1. The distillation framework proposed in the paper, tailored for MLLMs, demonstrates advantages over direct training in compressing image tokens.
2. The effectiveness of the proposed approach is validated on multiple datasets.
3. The paper provides detailed training parameters, execution time, and machine resource specifications.
4. The paper's figures are clearly rendered, effectively conveying the key concepts and enhancing readability.

Weaknesses:
1. The paper proposes a potential engineering implementation of distillation for multimodal models, without broader generalization or substantial methodological innovation: LCD based on the previous experimental finding that token compression in deeper layers of MLLMs is more difficult; TCD does not seem to be strongly related to MLLMs, but rather a general strategy for adapting distillation and compression. To the best of my knowledge, similar ideas have been proposed in multiple papers, such as 《Data-Free Knowledge Distillation For Image Super-Resolution》[1].
2. I think the derivation in the theoretical part is based on several preconditions and assumptions, leaving some room for improvement in terms of rigor. Therefore, the validation in the ablation experiment section is particularly critical, which is also what I focus on.  The experimental setup in the ablation study section is not clearly described. For the direct training without distillation loss, does it use the same dataset and the same training approach? Regarding the configuration without progressive compression, which layers are set? We understand that the selection of layers will affect the effect after compression. Have appropriate and consistent layers been set to maintain the validity of the ablation experiments?
3. The formulae and variables in the methodology section are presented in a relatively dense manner. It might be beneficial to optimize the presentation by using a more structured approach, such as an algorithmic flow, to enhance readability.
4. The experiments are primarily centered around LLaVA-v1.5-7B. To validate the effectiveness of the method more comprehensively, it is advisable to include additional models with varying architectures and parameter scales for evaluation. This would help demonstrate the generalizability of the approach across different model configurations.
5. The detailed descriptions of datasets and models in the appendix occupy a relatively large proportion of the content. It might be necessary to further optimize the presentation to better highlight the core findings and key details.

Reference:
- [1] Zhang, Yiman, et al. "Data-free knowledge distillation for image super-resolution." Proceedings of the IEEE/CVF Conference on Computer Vision and Pattern Recognition. 2021.

---

> ### Author Rebuttal · Authors · 2025-07-31
>
> Dear Reviewer BQCk,
>
> Thank you for your insightful feedback. We appreciate your thoughtful comments and are pleased to address each of your concerns in detail below.
>
> > **W1: Similar ideas in DFKD[1]**.
>
> **RW1:** Thank you for the thoughtful feedback. We appreciate the opportunity to clarify the distinctions between EPIC and prior KD methods. While there are conceptual connections to data-free KD methods such as the one proposed for image super-resolution (DFKD[1]), we would like to highlight some key methodological and architectural differences that set EPIC apart.
>
> 1. **Model Architecture & Modality**
>    - DFKD for SR employs a **GAN-based generator** to synthesize low-resolution images for a **CNN super-resolution network**.
>    - EPIC operates on **Transformer-based MLLMs**, where vision and language tokens interact via cross-modal attention.
> 2. **Task Nature**
>    - SR is a **pixel-level regression task** from LR → HR images.
>    - EPIC targets **multi-modal understanding**, where visual tokens must retain semantic fidelity **within a language modeling objective**—a fundamentally different optimization landscape.
> 3. **Distillation Paradigm**
>    - Traditional KD transfers knowledge from teacher to **separate student networks**.
>    - EPIC introduces **weight-sharing consistency distillation**: the same mllm acts as both teacher and student, eliminating the need for extra parameters or model replicas.
> 4. **Progressive Strategy Design**
>    - While both EPIC and DFKD adopt progressive mechanisms, their motivation and implementation are fundamentally distinct. DFKD's progressive strategy focuses on stage-wise expansion of student CNNs, suitable for pixel-level SR tasks. EPIC introduces a progressive mechanism tailored for MLLMs, which gradually adjusts (i) token compression ratios and (ii) the token compression layer to construct a progressive learning trajectory. This mechanism is designed to ensure training stability and semantic preservation, which, to the best of our knowledge, has not appeared in prior distillation literature, including DFKD.
> 5. **Theoretical Contribution**
>    - We provide the **first formal analysis** that progressive consistency paths yield lower total variation under convexity, specifically derived for **Transformer token compression**—a result inapplicable to GAN-CNN pipelines.
>
> We will include a comparative discussion with DFKD in the revised manuscript to clarify these aspects and hope our response is helpful.
>
> > **W2&Q1: Theory can be improved and detailed ablation setup**
>
> **RW2&Q1:**
> - **Theoretical part:** We believe those assumptions are standard in convex optimization literature and very common for theoretical tractability[2]. Those assumptions are essential to our theory, and we could explore how to safely remove them in our future work.  Also we believe our theory serves as an intuitive motivation and shows that progressive curriculum learning makes a smoother optimization trajectory (Line 153-156) and makes our work rigorously grounded.
> - **Detailed ablation setup:** We sincerely apologize for any confusion caused. The experiment without the distillation loss was trained using the same LLaVA-665K data, with all other training parameters and procedures kept identical to those of EPIC. For the variants without progressive compression ratio or progressive compression layer, we fixed the second layer as the compression layer. Overall, our ablation studies were conducted under a strictly fair and consistent experimental setup. We will include these details in the revised manuscript, and we would be happy to add any further information if needed.
>
> > **W3: Clarify method via structured/algorithmic presentation.**
>
> **RW3:** Thank you sincerely for your valuable suggestion. We will include an algorithmic flow and provide a more detailed methodological diagram in the revised version to enhance clarity and presentation.
>
> > **W4&Q2&Q5: More results on different models across varying parameter scales.**
>
> **RW4&Q2&Q5:** Thank you for this insightful suggestion. We have conducted experiments on LLaVA-v1.5-13B (a larger MLLM), Qwen-VL series (different model architecture). All experiments are trained on the same LLaVA-665K and have the same training configuration. As shown in the table below, **even when 77.8% of visual tokens are pruned, our approach retains over 95% of the average performance of the vanilla model, and with a 66.7% pruning ratio, it achieves performance on par with the vanilla model. Under the same pruning ratio, our method achieves up to a 4.9% higher average performance compared to direct SFT.**. These results demonstrate the effectiveness of EPIC across different MLLMs and parameter scales and we will include these results in the revised manuscript.
>
> |                   Model                   |  MME  | POPE  |  GQA  | TextVQA |  SQA  | MMbench_en | OCRBench | VizWiz | VQA_v2 |    Average    |
> | :---------------------------------------: | :---: | :---: | :---: | :-----: | :---: | :--------: | :------: | :----: | :----: | :-----------: |
> |               Qwen2.5-VL-3B               | 1738  | 86.6  | 62.5  |  58.4   | 74.2  |    66.8    |   315    |  48.7  |  72.0  |     62.5    |
> | + Direct SFT (Pruning Ratio 0.667) | 1702 | 82.3 | 60.1 | 55.8 | 73.1 | 64.8 | 300 | 48.1 | 69.3 | 60.5 (96.8%) |
> | + TCD (Pruning Ratio 0.667) | 1732  | 85.1  | 62.0  |  57.1   | 74.1  |    66.8    |   301    |  48.3  |  71.3  | 61.9 (99.0%) |
> | + TCD (Pruning Ratio 0.778) | 1711  | 83.7  | 61.4  |  55.6   | 73.5  |    65.3    |   287    |  47.6  |  70.1  | 60.8 (96.9%) |
> ||
> |                Qwen2-VL-7B                | 2212  | 85.9  | 67.3  |  80.1   | 82.1  |    79.2    |   664    |  53.6  |  77.9  |     74.6      |
> | + Direct SFT (Pruning Ratio 0.667) | 2110 | 82.8 | 66.1 | 77.3 | 80.1 | 78.5 | 649 | 52.0 | 75.9 | 72.6 (97.3%) |
> | + TCD (Pruning Ratio 0.667) | 2199  | 84.2  | 67.3  |  78.9   | 82.3  |    80.1    |   657    |  53.7  |  78.0  | 74.3 (99.6%)  |
> | + TCD (Pruning Ratio 0.778) | 2157  | 83.8  | 65.6  |  78.1   | 81.6  |    78.7    |   647    |  52.3  |  76.5  | 73.1 (98.0%)  |
> ||
> |              LLaVA-v1.5-13B               | 1850  | 86.1  | 62.3  |  61.5   | 72.3  |    67.2    |   332    |  57.5  |  74.7  |     52.8      |
> | + Direct SFT (Pruning Ratio 0.667) | 1715 | 84.2 | 61.4 | 66.0 | 67.3 | 65.3 | 518 | 61.1 | 73.2 | 65.7 (96.9%) |
> | + TCD (Pruning Ratio 0.667) | 1835  | 84.9  | 62.4  |  60.5   | 72.7  |    67.6    |   328    |  58.5  |  74.4  | 52.7 (99.8%)  |
> | + TCD (Pruning Ratio 0.778) | 1800  | 83.4  | 61.6  |  59.1   | 72.8  |    67.3    |   300    |  58.4  |  71.2  | 51.6 (97.7%)  |
>
> **(Due to space limits, more results on LLaVA-Next and Qwen2-VL-0.5B can be found in RW1 to Reviewer aYx1, or we can share the results in the discussion thread upon request. Thanks for understanding)**
>
> > **W5: Dataset/model appendix length issue**
>
> Thank you for your feedback. In the revised version, we will simplify and relocate the dataset and model descriptions to better highlight our key findings and essential details.
>
> > **Q3: Why some compressed models outperform original models**
>
> We appreciate the insightful question. The observation that compressed visual token models occasionally outperform their original counterparts is not an isolated phenomenon, but one supported by several plausible mechanisms:
> - **Reduced redundancy improves focus**: Compression removes redundant tokens (e.g., background, textures), preserving semantically salient regions and helping the model attend to key visual cues.
> - **Implicit regularization**:Excessive visual tokens may introduce noise or task-irrelevant details, leading to overfitting on non-generalizable patterns during training. Compression acts as an implicit regularizer, forcing the model to focus on more robust and generalizable features.
> - **Optimization benefits from the progressive training strategy (EPIC)**: Our method employs progressive consistency distillation, gradually guiding MLLM to adapt to compressed visual representations. This "from-easy-to-hard" learning paradigm facilitates smoother optimization and convergence to better minima.
> - **Task-specific needs**: Many tasks require only a few critical regions; excessive tokens can distract, harming performance.
>
> > **Q4: Why isn't the combined method included in Table 1?**
>
> **RQ4:** We sincerely thank you for raising this important point. The reason we did not include the combined version of TCD and LCD—termed *Integrated Progressive Consistency Distillation* (ICD)—in Table 1 is due to the following considerations, rather than any suboptimal performance of the combined approach:
> - **ICD results are fully reported**: The performance of ICD is presented in **Table 7 of Appendix D**. Notably, with only 192 tokens retained (a 66.7% reduction), ICD achieves an average score of 61.7% across 10 benchmarks—surpassing the vanilla LLaVA (61.4%)—demonstrating its effectiveness.
> - **Table 1 is designed for fair comparison with existing methods**:The primary purpose of Table 1 is to compare our method with existing token-compression approaches, most of which employ only a single compression strategy. To ensure a fair and consistent comparison, we present results for TCD-only and LCD-only, demonstrating the effectiveness of each strategy and highlighting the general applicability of the EPIC framework.
> - **Space and clarity:** Table 1 already includes 10 benchmarks and over 10 methods; adding ICD would increase clutter. To maintain clarity, we present the core components (TCD and LCD) in the main text and defer ICD—an integrated variant—to the appendix for a logical progression.
>
> **Reference**
>
> [1] Zhang, Yiman, et al. "Data-free knowledge distillation for image super-resolution." Proceedings of the IEEE/CVF Conference on Computer Vision and Pattern Recognition. 2021. \
> [2] Boyd, Stephen P., and Lieven Vandenberghe. Convex optimization. Cambridge university press, 2004.
>
> Best Regards,
>
> The Authors

---

> > ### Comment · Reviewer_BQCk · 2025-08-04
> >
> > Thank you for the authors' response. I still hold my initial view on the novelty of the paper. However, the authors have supplemented the ablation experiments with results from multiple different models, so I will consider raise my rating.

---

> > > ### Author Response · Authors · 2025-08-05
> > >
> > > Dear Reviewer BQCk,
> > >
> > > Thank you for your positive feedback. We are glad to know that our rebuttal has addressed most of your concerns, and we sincerely appreciate your willingness to raise the score.
> > >
> > > Regarding your concern about the novelty, we would like to further clarify: Existing methods like DFKD and traditional KD assume that the input remains complete or the data distribution stays unchanged. However, EPIC specifically tackles the challenge posed by dynamic token compression, which induces feature space perturbations (different scenarios and challenges). To address this, we introduced progressive learning strategy, with token-wise and layer-wise consistency distillation (tailored for the current challenge), along with weight-sharing distillation to avoid introducing additional parameters. These two innovations, specifically designed for MLLMs and token compression, have not been explored in previous works.
> > >
> > > Once again, thank you for your valuable suggestions and for considering raising the score. We hope this clarification helps in further understanding the contributions our work makes, and we will clarify these differences in the revised version.
> > >
> > > Best regards,
> > >
> > > The Authors

---

### Official Review · Reviewer_aYx1 · 2025-07-04

**Clarity:** 3
**Significance:** 3
**Originality:** 3
**Rating:** 4
**Confidence:** 3

**Summary:**

The paper introduces EPIC (Progressive Consistency Distillation), a training framework that makes multi-modal LLMs (MLLMs) more efficient by progressively compressing vision tokens. EPIC decomposes the perturbation caused by token pruning along two axes: (1) Token Consistency Distillation (TCD) gradually increases the compression ratio during finetuning, with a weight-sharing “teacher” that always keeps ≈ 5–10 pp fewer tokens than the “student”. (2) Layer Consistency Distillation (LCD) begins compressing only in the deepest Transformer layer and then shifts the compression layer toward shallower layers as training progresses.

**Questions:**

What is the wall-clock and A100-hour overhead of the extra teacher forward pass versus direct finetuning? A cost-accuracy plot would clarify practicality.

**Ethical Concerns:**

["NO or VERY MINOR ethics concerns only"]

**Final Justification:**

The author addressed my concerns. I am happy to maintain my score.

**Limitations:**

yes

**Quality:**

3

**Strengths And Weaknesses:**

Strengths

- Clear algorithm description and theoretical intuition.
- Comprehensive benchmark (10 vision–language tasks) and ablations on teacher loss & schedules.
- Shows inference-time gains (KV cache ↓ 89 %, FLOPs ↓ 84 %, latency 1.6×).
- Demonstrates that moderate token reduction (64-128) is a better speed/accuracy trade-off than extreme compression.

Weaknesses

- All experiments use a single 7 B LLaVA backbone. It can be beneficial to add larger MLLMs.
- Efficiency measured on POPE only; no wall-clock comparison against state-of-the-art token-reduction baselines (TokenPacker, MQT-LLaVA) with identical hardware.

---

> ### Author Rebuttal · Authors · 2025-07-31
>
> Dear Reviewer aYx1,
>
> Thank you for your insightful feedback. We appreciate your thoughtful comments and are pleased to address each of your concerns in detail below.
>
> > **W1: Add larger MLLMs.**
>
> **RW1:** Thank you for great suggestion. Evaluating across different parameter scales can further validate the effectiveness of EPIC. We have evaluated our method across MLLMs with varying architectures and sizes. All experiments are trained on the same LLaVA-665K SFT dataset and use the same experimental configuration as in Table 1. As shown in the table below, our method demonstrates strong performance across different parameter scales and diverse model architectures. **Even when 77.8% of visual tokens are pruned, our approach retains over 95% of the average performance of the vanilla model, and with a 66.7% pruning ratio, it achieves performance on par with the vanilla model. Under the same pruning ratio, our method achieves up to a 4.9% higher average performance compared to direct SFT.** We will include these results in the revised version.
>
> |                   Model                   |  MME  | POPE  |  GQA  | TextVQA |  SQA  | MMbench_en | OCRBench | VizWiz | VQA_v2 |    Average    |
> | :---------------------------------------: | :---: | :---: | :---: | :-----: | :---: | :--------: | :------: | :----: | :----: | :-----------: |
> |               Qwen2-VL-0.5B               | 1398  | 84.6  | 55.8  |  45.2   | 60.1  |    42.6    |   246    |  35.9  |  60.7  |     51.0      |
> | + Direct SFT (Pruning Ratio 0.667) | 1272 | 80.2 | 54.3 | 42.6 | 58.7 | 41.1 | 208 | 32.5 | 56.3 | 48.0 (94.1%) |
> | + TCD (Pruning Ratio 0.667) | 1379  | 83.1  | 56.1  |  44.7   | 61.3  |    42.0    |   231    |  34.8  |  59.8  | 50.5 (99.0%)  |
> | + TCD (Pruning Ratio 0.778) | 1375  | 80.0  | 55.2  |  41.1   | 61.2  |    41.6    |   207    |  34.1  |  56.7  | 48.9 (95.9%)  |
> ||
> |               Qwen2.5-VL-3B               | 1738  | 86.6  | 62.5  |  58.4   | 74.2  |    66.8    |   315    |  48.7  |  72.0  |     62.5     |
> | + Direct SFT (Pruning Ratio 0.667) | 1702 | 82.3 | 60.1 | 55.8 | 73.1 | 64.8 | 300 | 48.1 | 69.3 | 60.5 (96.8%) |
> | + TCD (Pruning Ratio 0.667) | 1732  | 85.1  | 62.0  |  57.1   | 74.1  |    66.8    |   301    |  48.3  |  71.3  | 61.85 (99.0%) |
> | + TCD (Pruning Ratio 0.778) | 1711  | 83.7  | 61.4  |  55.6   | 73.5  |    65.3    |   287    |  47.6  |  70.1  | 60.78 (96.9%) |
> ||
> |                Qwen2-VL-7B                | 2212  | 85.9  | 67.3  |  80.1   | 82.1  |    79.2    |   664    |  53.6  |  77.9  |     74.6      |
> | + Direct SFT (Pruning Ratio 0.667) | 2110 | 82.8 | 66.1 | 77.3 | 80.1 | 78.5 | 649 | 52.0 | 75.9 | 72.6 (97.3%) |
> | + TCD (Pruning Ratio 0.667) | 2199  | 84.2  | 67.3  |  78.9   | 82.3  |    80.1    |   657    |  53.7  |  78.0  | 74.3 (99.6%)  |
> | + TCD (Pruning Ratio 0.778) | 2157  | 83.8  | 65.6  |  78.1   | 81.6  |    78.7    |   647    |  52.3  |  76.5  | 73.1 (98.0%)  |
> ||
> |              LLaVA-v1.5-13B               | 1850  | 86.1  | 62.3  |  61.5   | 72.3  |    67.2    |   332    |  57.5  |  74.7  |     52.8      |
> | + Direct SFT (Pruning Ratio 0.667) | 1778 | 80.9 | 60.1 | 58.2 | 71.9 | 65.2 | 306 | 56.4 | 73.1 | 50.9 (96.4%) |
> | + TCD (Pruning Ratio 0.667) | 1835  | 84.9  | 62.4  |  60.5   | 72.7  |    67.6    |   328    |  58.5  |  74.4  | 52.7 (99.8%)  |
> | + TCD (Pruning Ratio 0.778) | 1800  | 83.4  | 61.6  |  59.1   | 72.8  |    67.3    |   300    |  58.4  |  71.2  | 51.6 (97.7%)  |
> ||
> |               LLaVA-Next-7B               | 1783  | 87.1  | 63.6  |  67.5   | 68.7  |    67.6    |   540    |  63.1  |  74.7  |     67.8      |
> | + Direct SFT (Pruning Ratio 0.667) | 1715 | 84.2 | 61.4 | 66.0 | 67.3 | 65.3 | 518 | 61.1 | 73.2 | 65.7 (96.9%) |
> | + TCD (Pruning Ratio 0.667) | 1809  | 85.9  | 63.1  |  66.9   | 68.7  |    68.7    |   537    |  62.7  |  73.4  | 67.5 (99.6%)  |
> | + TCD (Pruning Ratio 0.778) | 1740  | 85.4  | 62.2  |  65.1   | 68.3  |    67.9    |   523    |  60.9  |  74.4  | 66.5 (98.1%)  |
>
> > **W2: More wall-clock comparison**
>
> **RW2:** Thank you for the valuable suggestion. We have added inference efficiency evaluations across more benchmarks and conducted fair CUDA-time comparisons with state-of-the-art baselines, all under identical single A100 GPU hardware settings. As shown in table below, our three token compression strategies achieve inference latency comparable to TokenPacker and MQT-LLaVA when retaining **64 visual tokens**. In some cases (e.g., Random), our methods even outperform them in speed due to the absence of additional computation overhead. \
> Furthermore, we would like to emphasize that the central contribution of our work lies in the EPIC progressive learning framework, whose another key advantage is in training-time efficiency. As shown in Table 5 of the submission, our method achieves faster training compared to all baselines. The inference-time speedup primarily depends on the integrated token compression strategy, and our framework is modular and compatible with a wide range of such methods. In summary, our approach achieves competitive inference efficiency while significantly improving the training pipeline. We will include these results in the revised version.
>
> |       Benchmarks        | MME (2374 samples) | textvqa (5000 samples) | POPE (8910 samples) | MMB (4377 samples) |
> | :---------------------: | :----------------: | :--------------------: | :-----------------: | :----------------: |
> |  Vanilla LLaVA-v1.5-7B  |       300.7s       |         977.6s         |       1103.5s       |       772.6s       |
> |    DART     |       202.1s       |         712.8s         |       744.3s        |       436.3s       |
> |    FastV    |       211.0s       |         713.5s         |       749.1s        |       436.9s       |
> |   Random    |       189.7s       |         705.8s         |       697.3s        |       420.7s       |
> ||
> |  MQT-LLaVA  |       207.8s       |         720.1s         |       740.6s        |       430.4s       |
> | TokenPacker |       199.1s       |         710.7s         |       700.9s        |       422.6s       |
>
> > **Q1: A cost-accuracy plot on direct SFT and ours method.**
>
> **RQ1:** Thank you for your suggestion. Due to policy limits, we present the data for the cost-accuracy plot in the table below. We trained both LLaVA-v1.5-7B with direct SFT and our method—using the same LLaVA-665K—on a setup with 8×A100 GPUs, saving checkpoints at multiple training steps and record corresponding time. The results show that EPIC incurs a moderate and acceptable increase in training time. **Remarkably, the model trained with EPIC achieves performance on par with, or even surpassing, the vanilla model, despite removing 66.7% of the visual tokens during inference.** This outcome is indeed encouraging, demonstrating that high multimodal representation efficiency can be achieved without sacrificing accuracy. We will present the above data in a cost-accuracy plot in the revised version. Thank you once again.
>
> |                 Method                  | Training Steps | Wall-clock Time |  MME   | POPE  |  GQA  | TextVQA |  SQA  | MMbench_en | MMbench_cn | OCRBench | VizWiz | SEEDBench-2 | VQA-v2 | Average Score |
> | :-------------------------------------: | :------------: | :-------------: | :----: | :---: | :---: | :-----: | :---: | :--------: | :--------: | :------: | :----: | :---------: | :----: | :-----------: |
> | LLaVA v1.5 7B |      1000      |     1:55:36     | 1770 | 86.0  | 57.0  |  53.1   | 67.8  |    57.7    |    50.2    |  305   |  57.0  |    44.2     |  67.4  |     57.7     |
> | + Direct SFT (Pruning Ratio 0.667) | 1000 | 1:55:36 | 1734 | 84.1 | 55.7 | 50.2 | 66.2 | 55.6 | 48.6 | 289 | 55.8 | 43.4 | 65.9 | 56.0 |
> | + TCD (Pruning Ratio 0.667)    |      1000      |     2:22:05     | 1754 | 85.4  | 56.7  |  53.2   | 67.5  |    57.0    |    49.4    |  306   |  57.1  |    43.3     |  66.7  |     57.2     |
> ||
> | LLaVA v1.5 7B |      2000      |     3:54:47     | 1744 | 84.9  | 59.4  |  55.3   | 68.5  |    58.4    |    49.1    |  313   |  44.2  |    45.9     |  69.8  |     57.2     |
> | + Direct SFT (Pruning Ratio 0.667) | 2000 | 3:54:47 | 1712 | 81.0 | 57.9 | 53.1 | 66.9 | 57.2 | 47.8 | 301 | 43.1 | 43.9 | 67.3 | 55.4 |
> | + TCD (Pruning Ratio 0.667)     |      2000      |     4:46:32     | 1738 | 84.3  | 58.3  |  55.0   | 68.6  |    60.0    |    48.9    |  311   |  51.2  |    45.1     |  68.1  |     57.5     |
> ||
> | LLaVA v1.5 7B |      3000      |     5:51:42     | 1720 | 85.8  | 60.7  |  56.5   | 67.5  |    63.2    |    55.6    |  313   |  55.1  |    46.8     |  70.1  |     59.5     |
> | + Direct SFT (Pruning Ratio 0.667) | 3000 | 5:51:42 | 1698 | 84.3 | 59.0 | 54.9 | 66.1 | 62.3 | 54.1 | 299 | 53.5 | 45.2 | 69.2 | 58.1 |
> | + TCD (Pruning Ratio 0.667)     |      3000      |     7:22:53     | 1733 | 85.7  | 60.6  |  56.0   | 67.8  |    62.5    |    55.2    |  318   |  57.0  |    46.4     |  70.5  |     59.6     |
> ||                |
> | LLaVA v1.5 7B |      4000      |     7:49:30     | 1750 | 85.9  | 62.2  |  57.8   | 68.4  |    64.4    |    55.8    |  317   |  54.5  |    47.0     |  71.7  |     60.2     |
> | + Direct SFT (Pruning Ratio 0.667) | 4000 | 7:49:30 | 1721 | 83.2 | 60.3 | 56.3 | 66.9 | 63.2 | 54.8 | 306 | 52.6 | 46.2 | 69.3 | 58.6 |
> | + TCD (Pruning Ratio 0.667)     |      4000      |     9:43:38     | 1793 | 84.7  | 60.8  |  56.9   | 68.8  |    64.9    |    56.1    |  314   |  56.0  |    45.9     |  72.2  |     60.2     |
> ||
> | LLaVA v1.5 7B |      5000      |    10:43:29     | 1775 | 86.0  | 62.5  |  58.1   | 68.5  |    64.5    |    56.3    |  317   |  53.4  |    47.1     |  72.3  |     60.4     |
> | + Direct SFT (Pruning Ratio 0.667) | 5000 | 10:43:29 | 1720 | 84.3 | 60.6 | 57.9 | 66.4 | 63.5 | 54.8 | 305 | 52.5 | 46.0 | 70.2 | 58.9 |
> | + TCD (Pruning Ratio 0.667)     |      5000      |    11:58:44     | 1800 | 84.4  | 61.2  |  57.6   | 68.9  |    65.4    |    56.3    |  309   |  56.6  |    46.4     |  72.2  |     60.4     |
>
>
> Best Regards,
>
> The Authors

---

### Official Review · Reviewer_mKBF · 2025-07-11

**Clarity:** 3
**Significance:** 3
**Originality:** 3
**Rating:** 4
**Confidence:** 3

**Summary:**

This paper tackles the problem of efficient compression of visual tokens in MLLMs. The authors propose a new progressive training-aware token compression that enables various "scales" of compression: token level or layer level. The main principle could be considered as a curriculum learning strategy with easy learning tasks with a low token compression ratio at the beginning of the training and more difficult ones after. Their approach is based on a teacher-student architecture with a progressive consistency distillation. The paper contains a theoretical motivation and a formalization of the proposed approach. An experimental validation is provided on 10 visual understanding benchmarks. Experimental results show the interest of the approach. It also includes ablation studies.

**Questions:**

+ The authors propose to use a linear increase of the compression ratio in the EPIC framework. The choice is relevant, but alternative progressive schemes could also be proposed. For instance, recent approaches have been proposed for learning a curriculum temperature of KD distillation that could be relevant. How the proposed approach could be enriched with other curriculum strategies?

**Ethical Concerns:**

["NO or VERY MINOR ethics concerns only"]

**Limitations:**

Yes

**Paper Formatting Concerns:**

+ The colors of Fig 1 could be changed. Indeed, the chosen colors made it unreadable.

**Quality:**

3

**Strengths And Weaknesses:**

** Strengths**

+ The progressive distillation principle and associated framework are relevant and interesting, with an experimental validation that supports the main claim of the paper.
+ The multi-scale distillation (token and layer-wise) is also a nice idea.
+ The proposed framework enables the easy integration of  plug-and-play token compression strategies
+ The paper is well written with a clear formalization of the proposed approach
+ The attempt to provide a theoretical intuition of the progressive consistency distillation is a positive point of the paper.
+ Experimental part supports the claims of the paper.

**Weaknesses**

+ Some cognitive motivations for the proposed approach are given without any justifications. For instance, it is not clear to me how the D. Marr works enable us to argue that LLMs only need a small amount of information-dense text tokens.
+While the motivation is to improve the efficiency of compression of visual tokens in MLLMs, it's not clear to me to what extent the proposed approach is specific to this type of model. Couldn't it be generalized to purely visual models or tokens of other modalities? It's not clear to me to what extent the proposed approach is specific to this type of model. Couldn't it be generalized to purely visual models or tokens of other modalities?

---

> ### Author Rebuttal · Authors · 2025-07-31
>
> Dear Reviewer mKBF,
>
> Thank you for your insightful and constructive feedback. We sincerely appreciate your thoughtful comments and are pleased to address each of your concerns in detail below.
>
> > **W1: Some cognitive motivations for the proposed approach are given without any justifications. For instance, it is not clear to me how the D. Marr works enable us to argue that LLMs only need a small amount of information-dense text tokens.**
>
> **RW1:** We sincerely apologize for any confusion this may have caused. In fact, our reference to D. Marr’s work[1] aims to highlight the higher information density of text tokens compared to visual tokens. This distinction is clearly emphasized in his research. Unlike LLMs, MLLMs must process not only text tokens but also a significantly larger number of visual tokens derived from images or videos. Therefore, we believe it is reasonable to argue that LLMs only require a small number of information-dense text tokens in most scenarios. We will clarify this connection between D.Marr’s work and our motivation in the revised version.
>
> > **W2: While the motivation is to improve the efficiency of compression of visual tokens in MLLMs, it's not clear to me to what extent the proposed approach is specific to this type of model. Couldn't it be generalized to purely visual models or tokens of other modalities?**
>
> **RW2:** Thank you for this insightful question. We focus on MLLMs because visual token redundancy is especially critical in high-resolution and long-video scenarios, directly affecting performance and user experience. That said, our compression framework is not limited to MLLMs. It can be extended to pure vision models, such as ViTs, by applying token reduction within the visual encoder under our training paradigm. Beyond vision, our approach could extend to other modalities. For example, CoT tokens in reasoning models often exhibit redundancy, and applying targeted compression strategies within a compatible training framework could enhance efficiency in language-centric models as well.
> We conducted experimental validation by integrating the token compression operation into the purely visual model—ViT—within the EPIC framework, updating the ViT weights, while keeping all other experimental settings identical to those in Table 1. The token pruning strategy for training and evaluation is DART.
> As shown in the table, applying our method within the visual model (ViT) still achieves strong performance, attaining 99.5% of the average score of the vanilla model. This demonstrates the broad applicability and effectiveness of our approach.
>
> |                         Model                         |  MME  | POPE  |  GQA  | TextVQA |  SQA  | MMbench_en | MMbench_cn | OCRBench | VizWiz | VQA_v2 | Average |
> | :---------------------------------------------------: | :---: | :---: | :---: | :-----: | :---: | :--------: | :--------: | :------: | :----: | :----: | :-----: |
> |                     LLaVA v1.5 7B                     | 1785  | 85.9  | 61.9  |  58.1   | 68.3  |    64.1    |    55.8    |   319    |  52.5  |  72.2  |  61.4   |
> | + TCD (Pruning Ratio 0.778)     | 1861  | 84.5  | 59.9  |  56.6   | 70.8  |    65.6    |    54.3    |   299    |  54.9  |  69.7  |  61.3   |
> | + TCD in ViT (Pruning Ratio 0.778) | 1823  | 83.5  | 60.9  |  57.2   | 68.6  |    64.3    |    55.4    |   312    |  53.6  |  70.7  |  61.1   |
>
>
> We believe these are promising directions and plan to explore them in future work.
>
>
> > **Q1: The authors propose to use a linear increase of the compression ratio in the EPIC framework. The choice is relevant, but alternative progressive schemes could also be proposed. For instance, recent approaches have been proposed for learning a curriculum temperature of KD distillation that could be relevant. How the proposed approach could be enriched with other curriculum strategies?**
>
> **RQ1:** Thank you for the insightful suggestion. In our current work, we adopted a linear schedule due to its simplicity and demonstrated its effectiveness across multiple benchmarks. Our progressive learning strategy is specifically designed in conjunction with token compression, aiming to improve the efficiency of MLLMs by dynamically adjusting either the compression ratio or the compression layer throughout training. We also agree that richer curriculum strategies could further enhance training. The curriculum temperature technique you mentioned is a general curriculum learning approach used in KD. Importantly, it is complementary to our framework and can be seamlessly integrated to further guide student learning.  \
> We therefore followed [2] to incorporate curriculum temperature into EPIC, conducting experiments under the same setup as main table 1 (token pruning strategy: DART). As shown in the results below, the addition of curriculum temperature leads to performance improvements on certain benchmarks, yet the overall average performance exhibits only minor fluctuations compared to TCD. This suggests that simply stacking progressive learning strategies may not be sufficient to unlock their full potential—more sophisticated integration may be required. Alternatively, our method might already be approaching the performance upper bound under token compression, where curriculum temperature does exert an effect, but its benefits are not clearly reflected in the metrics.
>
> |                                  Model                                  |  MME  | POPE  |  GQA  | TextVQA |  SQA  | MMbench_en | MMbench_cn | OCRBench | VizWiz | VQA_v2 | Average |
> | :---------------------------------------------------------------------: | :---: | :---: | :---: | :-----: | :---: | :--------: | :--------: | :------: | :----: | :----: | :-----: |
> |                              LLaVA v1.5 7B                              | 1785  | 85.9  | 61.9  |  58.1   | 68.3  |    64.1    |    55.8    |   319    |  52.5  |  72.2  |  61.4   |
> |             + TCD (Pruning Ratio 0.778)              | 1861  | 84.5  | 59.9  |  56.6   | 70.8  |    65.6    |    54.3    |   299    |  54.9  |  69.7  |  61.3   |
> | + TCD&Curriculum Temperature (Pruning Ratio 0.778) | 1835  | 85.2  | 60.3  |  56.4   | 70.9  |    65.5    |    54.8    |   301    |  54.2  |  70.1  |  61.3   |
>
> We will include a discussion of this potential integration in the revised version and consider it a valuable direction for future work.
>
> **Reference**
>
> [1] Marr, David. Vision: A computational investigation into the human representation and processing of visual information. MIT press, 2010. \
> [2] Li, Zheng, et al. "Curriculum temperature for knowledge distillation." Proceedings of the AAAI Conference on Artificial Intelligence. Vol. 37. No. 2. 2023.
>
> Best Regards,
>
> The Authors

---

> > ### Author Response · Authors · 2025-08-08
> >
> > Dear Reviewer mKBF,
> >
> > Thank you again for your time and insightful feedback on our paper.
> >
> > As the discussion period is concluding soon, we just wanted to gently follow up. We hope our responses have clarified the concerns you raised. Please let us know if you have any further questions or if there is any other information we can provide.
> >
> > We look forward to hearing from you and appreciate your valuable suggestions for improving our work.
> >
> > Sincerely,
> >
> > The Authors

---

### Note · Authors · 2025-08-13

Dear SAC, AC, and Reviewers,

We sincerely thank you for your time and effort. We are pleased to receive positive feedback from the reviewers (mKBF, aYx1, BQCk, Guzs) in their initial comments and during the discussion. We appreciate the reviewers’ recognition of our work and are glad that our responses have addressed their concerns.

Below, we summarize the discussions with the reviewers and the revisions we have made:

- **Generalizability Across MLLMs (aYx1, BQCk, Guzs)**: To validate EPIC’s effectiveness, we conducted experiments on diverse architectures (LLaVA-v1.5-7B/13B, Qwen2-VL-0.5B/3B/7B, LLaVA-Next-7B) and scales. Results show EPIC retains >95% of vanilla performance with 77.8% token pruning and achieves up to 4.9% higher average performance than direct SFT at 66.7% token pruning, demonstrating strong effectiveness and generalizability.
- **Novelty Compared to Prior KD Methods (BQCk)**: We clarified EPIC’s distinctions from methods like DFKD, emphasizing its weight-sharing consistency distillation, modality-specific progressive strategies (token-wise and layer-wise), and differences in task nature and model architecture. These tailored designs address MLLM-specific challenges, setting EPIC apart from traditional KD.
- **Quantitative Support for Motivation (Guzs)**: To substantiate the qualitative illustration in Fig. 1, we compared direct training (fixed 60% random pruning over 4000 steps) with progressive learning (0%→60% pruning over 4000 steps) on LLaVA-v1.5-7B. Progressive learning outperformed direct training on average across 6 benchmarks (e.g., 60.8 vs. 58.9), validating the motivation behind Fig. 1.
- **Exploration in Pure Vision Models and Curriculum Temperature (mKBF)**: We applied EPIC to ViT, achieving 99.5% of the vanilla model’s performance with 77.8% pruning. Integrating curriculum temperature into EPIC yielded minor improvements (e.g., 61.3% average score vs. 61.3% for TCD alone), suggesting potential for further optimization, which we will explore in future work.

We commit to incorporating the above results and clarifications—along with enhanced algorithmic descriptions, comparative discussions, and sensitivity analyses (e.g., λ in [0.5, 0.7])—into the revised manuscript to improve clarity and rigor.

We thank the reviewers for their valuable feedback and believe that our method will bring further inspiration and practical insights to community, especially in designing and training efficient MLLMs.

Best regards,

The Authors

---

### Decision · Program_Chairs · 2025-09-17

**Decision:**

Accept (poster)

**Comment:**

The paper proposes a progressive distillation framework for compressing the vision tokens in MLLMs. The key insight is that naively training a MLLM with token compression leads to quality drop. To avoid this, the paper proposes to rely on co/self-distillation where multiple forward passes are made through the same model at different token compression rates and the joint loss is optimized. The information flow from the teacher helps smooth the optimization landscape and leads to better solutions. In addition to co-distillation, the paper also proposes a curriculum for teacher and student to make sure the teacher model is not too far from the student model in terms of complexity. Experimental results on Vicuna-7B, Qwen2-VL-7B show that the proposed technique achieves a better quality-efficiency tradeoff

While the empirical evidence looks promising, the paper does a poor job in citing related work, and positioning the paper. For instance, it is claimed that **EPIC introduces weight-sharing consistency distillation: the same mllm acts as both teacher and student, eliminating the need for extra parameters or model replicas.**

As **Reviewer BQCk** pointed out, this is inaccurate and doesn't properly credit prior works which introduced self distillation. For instance, [1] also performs weight sharing between teacher and student during training. There are multiple works which build on [1] and applied it to various settings [2]. And I believe the distillation community has many more works on this topic.  It is important to cite these works and compare against them either empirically or at a conceptual level.

While I'm recommending accepting the paper, I'd encourage the authors to properly position their work within the broader area of distillation.


[1] Matryoshka Representation Learning https://arxiv.org/abs/2205.13147
[2] Matryoshka Quantization https://arxiv.org/abs/2502.06786